# Importin-β modulates the permeability of the nuclear pore complex in a Ran-dependent manner

**Alan R Lowe**[1,2,3,4,5][*][†], **Jeffrey H Tang**[3,4,5,6,7,8,9][†], **Jaime Yassif**[3,4,5,9], **Michael Graf**[10], **William YC Huang**[11], **Jay T Groves**[4,5,9,11,12], **Karsten Weis**[7,8][*], **Jan T Liphardt**[3,4,5,6,7,9,12][*]

[1]Institute for Structural and Molecular Biology, University College London and Birkbeck College, London, United Kingdom; [2]London Centre for Nanotechnology, University College London and Birkbeck College, London, United Kingdom; [3]Department of Physics, University of California, Berkeley, Berkeley, United States; [4]QB3, University of California, Berkeley, Berkeley, United States; [5]Bay Area Physical Sciences Oncology Center, University of California, Berkeley, Berkeley, United States; [6]Department of Bioengineering, Stanford University, Stanford, United States; [7]Department of Molecular and Cell Biology, University of California, Berkeley, Berkeley, United States; [8]Institute of Biochemistry, Eidgenössische Technische Hochschule Zürich, Zürich, Switzerland; [9]Biophysics Graduate Group, University of California, Berkeley, Berkeley, United States; [10]Section of Life Sciences and Technologies, École polytechnique fédérale de Lausanne, Lausanne, Switzerland; [11]Department of Chemistry, Howard Hughes Medical Institute, University of California, Berkeley, Berkeley, United States; [12]Physical Biosciences Division, Lawrence Berkeley National Laboratory, Berkeley, United States

**\*For correspondence:** a.lowe@ucl.ac.uk (ARL); karsten.weis@bc.biol.ethz.ch (KW); jan.liphardt@stanford.edu (JTL)

[†]These authors contributed equally to this work

**Abstract** Soluble karyopherins of the importin-β (impβ) family use RanGTP to transport cargos directionally through the nuclear pore complex (NPC). Whether impβ or RanGTP regulate the permeability of the NPC itself has been unknown. In this study, we identify a stable pool of impβ at the NPC. A subpopulation of this pool is rapidly turned-over by RanGTP, likely at Nup153. Impβ, but not transportin-1 (TRN1), alters the pore's permeability in a Ran-dependent manner, suggesting that impβ is a functional component of the NPC. Upon reduction of Nup153 levels, inert cargos more readily equilibrate across the NPC yet active transport is impaired. When purified impβ or TRN1 are mixed with Nup153 in vitro, higher-order, multivalent complexes form. RanGTP dissolves the impβ•Nup153 complexes but not those of TRN1•Nup153. We propose that impβ and Nup153 interact at the NPC's nuclear face to form a Ran-regulated mesh that modulates NPC permeability.

## Introduction

The nuclear pore complex (NPC) is a very large cellular transport channel conserved among all eukaryotes. The NPC controls the nuclear entry and exit of cargos ranging from single proteins to large ribonucleoprotein complexes (*Stewart, 2007*; *Peters, 2009*). Cargos smaller than ~40 kDa can passively equilibrate across the nuclear envelope while larger cargos must bind special transport receptors to move from the cytoplasm into the nucleus and accumulate there (*Stewart, 2007*; *Peters, 2009*). These transport receptors are able to bind cargos but can also interact with unstructured phenylalanine–glycine repeat proteins (FG nucleoporins) within the pore. Directional transport of

**eLife digests** In our cells, genetic material is contained within the nucleus, which is separated from the rest of the cell by a double-layered membrane called the nuclear envelope. Within this membrane there are pores that allow proteins and other molecules to enter and exit the nucleus.

Small molecules can pass through these pores unaided, which is known as 'passive' transport. However, larger cargos need help from transport receptor proteins in a process called 'active' transport. Large cargos bind to transport receptors, such as importin-β, in the cytoplasm and are then guided through the pore. Once the cargo and importin-β are inside the nucleus, a protein called RanGTP binds to importin-β to release the cargo.

It is thought that importin-β and RanGTP are only important for the active transport of cargo. Here, Lowe et al. studied how importin-β interacts with the pore. The experiments show that in the absence of RanGTP, importin-β accumulates inside the pore and binds to a protein called Nup153, which is part of the complex of proteins that makes up the pore. However, when RanGTP is present, some of the importin-β is displaced from Nup153 and leaves the pore, which makes it easier for cargo to pass through.

Further experiments show that when Nup153 and importin-β are mixed, they associate into a gel-like material that can be 'melted' by RanGTP. Lowe et al. propose a model for how RanGTP may control the flow of cargo through the nuclear pore by affecting the binding of importin-β to Nup153. Lowe et al.'s findings suggest that passive and active transport of cargo across the nuclear pore are fundamentally connected and suggest that RanGTP provides the cell with an additional layer of control over nucleocytoplasmic transport.

cargos is powered by the small GTPase Ran and a system of compartment-specific GTP hydrolysis and GDP-to-GTP exchange, which establishes a sharp concentration gradient of RanGTP across the nuclear envelope (*Izaurralde et al., 1997*; *Kalab et al., 2002*). Importin-β (impβ) and other members of the karyopherin family of nuclear transport receptors form a complex with their cognate cargos in the RanGTP-low cytoplasmic environment and release cargos upon binding to RanGTP in the nucleus.

Contemporary transport models ('selective phase' hydrogel [*Ribbeck and Gorlich, 2001*; *Frey et al., 2006*], 'virtual gate' [*Rout et al., 2003*], 'reduction of dimensionality' [*Peters, 2005*], and 'polymer brush' [*Lim et al., 2007*]) address the behavior of FG nucleoporins and transport receptors to explain the NPC's selectivity and ability to facilitate cargo diffusion. A tacit implication of these models is that the diffusive movement of cargos through the NPC and the overall directionality of active transport are fundamentally distinct and separate processes. In this perspective, cargo–receptor complexes are expected to equilibrate freely across the nuclear envelope in the absence of an energy bias such as the RanGTP gradient, and efficient cargo accumulation against a concentration gradient requires only the Ran-driven unbinding of cargo molecules from their transport receptors.

However, several intersecting lines of evidence raise the possibility that RanGTP influences the permeability of the NPC itself, rather than only acting on cargos once they have completely entered the nuclear compartment. First, early studies have suggested that Ran is needed for cargos to move through the NPC (*Moore and Blobel, 1993*; *Gorlich et al., 1994*; *Moore and Blobel, 1994*). Second, it was shown that Ran plays an important role in dissociating impβ from the nuclear face of the NPC, in addition to displacing impβ from cargos (*Gorlich et al., 1996*). Third, extended tracking of cargos within single pores revealed a substantial RanGTP-dependent asymmetry in the cargo's exit step. Without RanGTP, cargos entered the pore but had an ∼100-fold higher probability of exiting the pore at the cytoplasmic face than the nuclear face, suggesting that RanGTP influences barriers felt by cargo-impβ complexes within the pore (*Lowe et al., 2010*). The Ran-sensitive exit asymmetry of large cargo-receptor complexes suggests that a currently unexplained Ran-dependent process takes place inside the channel near the nuclear face of the pore at about ∼70 nm along the transport axis (*Lowe et al., 2010*) (*Figure 1A*). Fourth, the overall transport success of large cargos appears to be more sensitive to RanGTP levels than other cargos (*Lyman et al., 2002*; *Snow et al., 2013*) and thus RanGTP somehow influences the interplay of cargo size and active transport. This latter connection is not necessarily mediated by cargo multivalency (*Snow et al., 2013*). Together, these observations hint

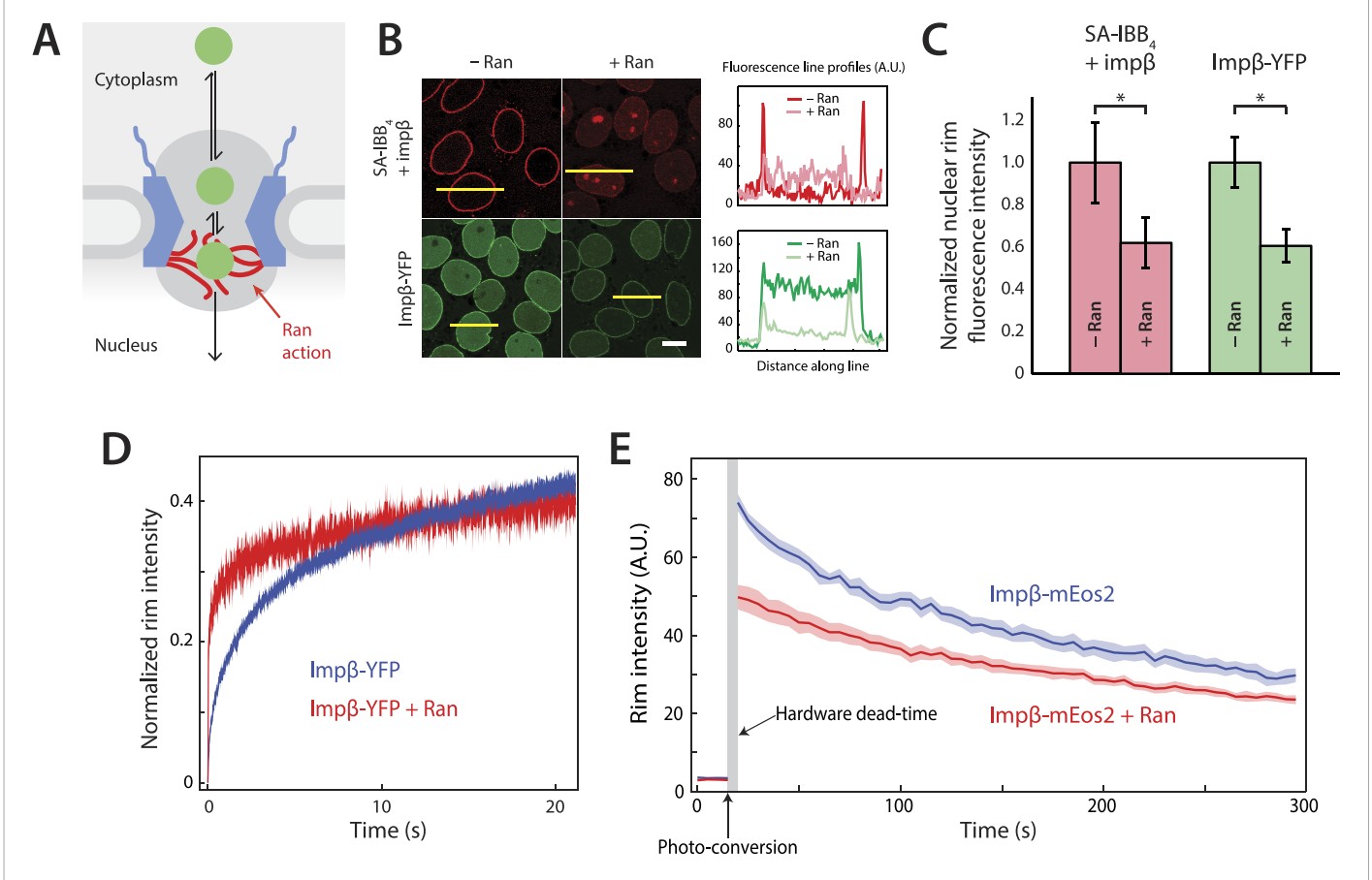

**Figure 1**. Effect of Ran on impβ binding affinity and turnover at the NPC. (**A**) Schematic of the NPC showing the location of the Ran-dependent exit step for cargo-receptor complexes. (**B**) Representative images of cargo-receptor complexes and impβ-YFP stalled within the pore and forming bright nuclear rims in the absence of Ran. Fluorescence intensity profiles are plotted for the yellow lines showing the nuclear rim intensity drop when Ran is added. Scale bar (white): 10 μm. (**C**) Nuclear rim fluorescence intensities of nuclei in (**B**) normalized to −Ran condition. Error bars represent the standard deviation about the mean. Asterisks (*) indicate a significant $p < 10–20$ using the Mann–Whitney U test (N ≥ 97 for all conditions). (**D**) Representative FRAP recovery curves of impβ-YFP at the nuclear envelope showing rapid initial recovery of impβ in the presence of Ran with evidence of a second, slower pool of impβ. Recovery in the absence of Ran (blue trace) is considerably slower. (**E**) Photo-conversion based characterization of the slowly dissociating impβ pool. Ran reduces the initial fluorescence signal but does not clear all impβ's from the NPC as shown by the residual fluorescence at the nuclear rim lasting hundreds of seconds. Shaded regions indicate the standard error of the mean (N = 20 for both conditions).

The following figure supplements are available for figure 1:

**Figure supplement 1**. Effect of impβ concentration on active transport.

**Figure supplement 2**. Schematic of the FRAP microscope.

**Figure supplement 3**. Photoconversion experiment details.

at additional Ran-driven processes within the pore that are not addressed by current models of active nucleocytoplasmic transport.

The spatial and temporal arrangement of factor(s) that allows the NPC to almost perfectly prevent large cargo translocation in the absence of RanGTP (error rate <1% [*Lowe et al., 2010*]) but allows efficient directional transport in its presence remains unknown. In this study, we investigate the interaction of impβ and Ran within single NPCs using quantitative biophysical measurements, and we relate those interactions to cargo translocation. We identify Nup153 as an important impβ binding partner at the nuclear face of the pore. The impβ•Nup153 interaction is Ran-sensitive and contributes

to the NPC permeability barrier in vitro. Ran's effect on impβ turnover, stoichiometry, and spatial distribution at the nuclear pore is characterized, and the impβ•Nup153 binding behavior is examined. We propose a mechanism for how Ran-dependent modulation of impβ at Nup153 may contribute to the NPC's selective permeability.

## Results

### Cargo-impβ complexes stall inside the NPC in the absence of Ran

We employed the commonly used in vitro nuclear transport assay (digitonin permeabilized HeLa cells supplemented with exogenous recombinant transport factors) (*Adam et al., 1990*; *Lowe et al., 2010*) to characterize impβ-mediated nuclear transport. This assay yields nuclei with functional NPCs while allowing us to control the composition and concentrations of transport factors and cargos. A model cargo consisting of a fluorescently labeled tetravalent streptavidin (SA) bound to biotinylated impβ binding (IBB) domains was used to examine cargo binding at the NPC. The streptavidin-IBB tetramer (SA-IBB$_4$) cargo is large (~218 kDa) and contains multiple import signals, as do many natural large cargos. In the presence of physiological levels of impβ (1 μM, *Figure 1—figure supplement 1*), SA-IBB$_4$ strongly stains the nuclear envelope but does not efficiently enter the nuclear interior (*Figure 1B, C*). This indicates that cargo molecules accumulate within the NPC, presumably due to their inability to complete their translocation into the nucleus in the absence of Ran. However, when RanGDP (5 μM) + GTP (2 mM) (henceforth referred to as RanGTP) is added, the fluorescence intensity of the nuclear rim drops while that of the nuclear interior increases, showing that the cargos then efficiently exit the NPC and accumulate in the nucleus.

To further characterize this RanGTP dependence, we fluorescently labeled impβ with a YFP tag, yielding impβ-YFP, and examined how this transport receptor binds the NPC in the presence or absence of RanGTP. As expected for a FG-binding karyopherin, impβ-YFP formed a bright nuclear rim when added without RanGTP. However, as with the SA-IBB$_4$ cargos, the impβ-YFP signal at the rim was substantially reduced but not completely eliminated by RanGTP (*Figure 1B,C*). Ran thus modulates the way in which the NPC interacts with both cargo-bound impβ and free impβ. Moreover, since RanGTP reduces the impβ-YFP rim fluorescence, at least a subset of non-cargo engaged but NPC-bound transport receptors must be RanGTP-sensitive.

### Evidence for two functionally distinct pools of impβ at the NPC

We used fluorescence recovery after photobleaching (FRAP) to characterize the turnover kinetics of impβ at the NPC and to examine the binding affinity of impβ for the pore. For the FRAP experiments, impβ-YFP was allowed to form a fluorescent rim at the nuclear envelope and a section of that rim was then photobleached (the custom hardware is described in *Figure 1—figure supplement 2*). We subsequently monitored the recovery of rim fluorescence in the photobleached region. In the absence of RanGTP, the initial recovery of the impβ signal after bleaching took several seconds (time to 20% recovery = 1.6 ± 0.1 s, N = 20, *Figure 1D*, blue). However, with RanGTP, the initial recovery was 16-fold more rapid (time to 20% recovery = 0.1 ± 0.1 s, N = 20, *Figure 1D*, red). Therefore, as already indicated by the simple rim fluorescence experiments, RanGTP is able to accelerate the cargo-independent turnover of impβ bound to the NPC.

Inspection of the recovery traces hinted at a long-lived population of NPC-bound impβ with little or no turnover. Consistent with this, previous single-molecule titration experiments have suggested the presence of two types of impβ binding sites within the NPC (*Tokunaga et al., 2008*). To directly observe the slow turnover impβ subpopulation, we used a two-color photo-conversion approach (*Figure 1—figure supplement 3*). The photo-conversion hardware and geometry was optimized for quantification of subpopulations with extremely slow or no turnover, at the expense of introducing a multi-second dead time (*Figure 1E*, arrow) immediately following photo-conversion.

In these photo-conversion experiments, impβ was tagged with the photo-convertible fluorescent protein mEos2 (impβ-mEos2), initially yielding a green signal at the nuclear envelope. A small portion of the rim was then photo-converted to a red state. As bound (red) molecules are replaced by fresh non-converted (green) impβ-mEos2 from solution, the red rim signal gradually fades and the green rim signal gradually recovers, revealing the dissociation rate of bound impβ transport receptors. This red-to-green replacement process can be quantified for long times with good signal-to-noise. In the absence of RanGTP, the photo-converted (red) impβ-mEos2 signal decayed to half its initial value

within 3–4 min (*Figure 1E*), showing that some impβ molecules remain at the pore for long times. In the presence of RanGTP, the initial red signal was lower than without RanGTP (~50 AU vs ~75 AU), consistent with the previously detected RanGTP-dependent rapid turnover of impβ within the pore. Strikingly, however, some impβ molecules remained at the pore for several minutes, even in the presence of RanGTP (*Figure 1E*).

Summarizing, the FRAP experiments allowed us to quantify fast reactions within the pore and the photo-conversion experiments permitted quantification of slow reactions within the pore. Together, these experiments suggest that there are at least two pools of impβ within the NPC. One pool is stably bound to the NPC for many minutes, with and without RanGTP. The other pool is stably bound to the NPC only in the absence of RanGTP.

## Super-resolution imaging of impβ's spatial distribution at the NPC

Having detected two kinetically distinct impβ pools within the pore, we sought to characterize their spatial distribution and identify the nucleoporins they were binding. We were especially interested in the RanGTP-sensitive impβ pool, since RanGTP drives active transport and RanGTP-induced alterations of pore organization could therefore be relevant to active transport. We directly imaged and localized individual Cy5 or Alexa647 dye-labeled impβ molecules within the pore using dSTORM super-resolution localization microscopy (*Heilemann et al., 2008*; *van de Linde et al., 2011*) (*Figure 2A–D*; mean spatial precision, $\sigma_{x,y}$ of 12 nm, *Figure 2—figure supplement 1*). By directly labeling impβ with a fluorescent reporter, we removed additional localization uncertainty error (commonly referred to as linkage error) associated with the antibody labeling methods normally used for super-resolution or electron microscopy.

The dSTORM images taken at the equatorial plane of the nucleus show discrete elongated structures oriented normal to the nuclear envelope (*Figure 2B,E*). Viewed from the bottom of the nucleus, we observe radially symmetric, punctate NPC structures (*Figure 2D*). We did not see a recently reported (*Ma et al., 2012*) 'ring'-like distribution of impβ, although this could be a consequence of our spatial precision.

To visualize the axial distribution of impβ, we developed approaches for comparing hundreds to thousands of individual NPCs. Individual NPCs were identified by calculating an 'envelope histogram' of the number of localizations in a window normal to the nuclear envelope path (*Figure 2E*). Well-separated peaks within this histogram, containing a threshold number of localizations, indicate the position of putative NPCs and were selected for further study. The impβ localizations belonging to these NPCs were then extracted, rotated according to the interpolated envelope normal vector (*Figure 2F*) and aligned along the transport axis. Those structures requiring very large alignment shifts, or having poor correlation with the remainder of the data set, were removed.

Having extracted, rotated, and aligned the NPCs, we averaged the impβ localizations (*Figure 3A*). Viewed along the transport axis, there are two pools of impβ localizations separated by approximately 90 nm and occupying a footprint and spatial arrangement consistent with structural studies (*Frenkiel-Krispin et al., 2010*). Antibody labeling of Nup358/RanBP2 (located on the NPC cytoplasmic filaments) with a second fluorescent dye was used to confirm that the outermost pool of the impβ signal spatially overlapped with the cytoplasmic face of the NPC (*Figure 3—figure supplement 2*). The central channel measured ~50 nm at the narrowest point, consistent with single quantum dot transport studies (*Lowe et al., 2010*) and other super-resolution measurements (*Loschberger et al., 2012*). The addition of RanGTP, which produces a transport-competent pore with active impβ turnover, markedly decreased the total number of impβ localizations (*Figure 3A*, 'RanGTP' and *Figure 3—figure supplements 1, 3*) but also changed the shape of the probability density function (PDF) of impβ molecules within the pore. In the presence of RanGTP, the PDF is bimodal and shows a depletion of impβ from the nucleoplasmic face of the NPC (*Figure 3C*, compare red trace to black trace). RanGTP was therefore not simply displacing impβ from the pore but was displacing impβ preferentially from specific sites within the pore.

Based on the dSTORM data and particle-tracking studies that suggest that the end of the channel is the functional site of Ran action (*Lowe et al., 2010*), we hypothesized that Nup153 might be a site of RanGTP-sensitive impβ binding. Nup153 is an important terminal binding site for the impβ transport pathway (*Shah et al., 1998*; *Walther et al., 2001*), can bind as many as seven impβ molecules (*Milles and Lemke, 2014*), and interacts with Ran (*Saitoh et al., 1996*; *Ball and Ullman, 2005*; *Schrader et al., 2008*). Since Nup153 is essential for cell viability, we used partial RNAi

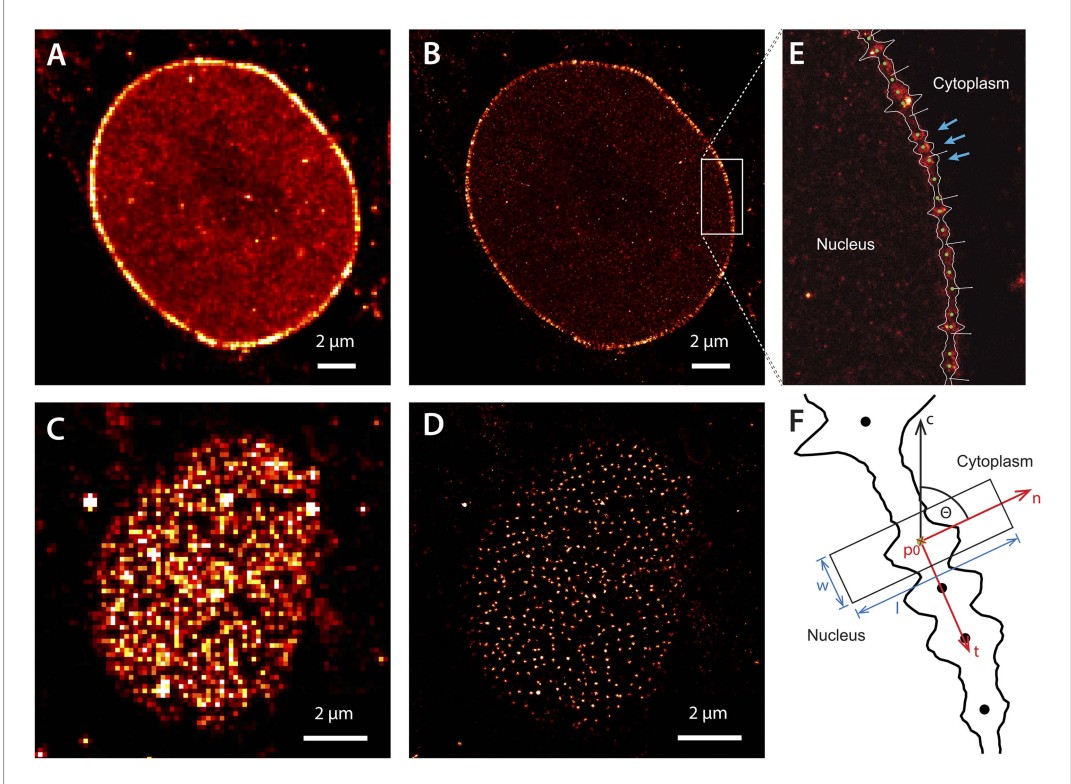

**Figure 2**. Super-resolution imaging of Alexa647-labeled impβ in digitonin-permeabilized HeLa cells. (**A**) Simulated widefield impβ localization at the equatorial plane of the nucleus and (**B**) corresponding dSTORM image. Mean localization precision is 12 nm. (**C** and **D**) Corresponding widefield and dSTORM images taken at the basal surface of the nucleus showing characteristic punctate NPC structures. (**E**) Zoom of the dSTORM image in (**B**), showing discrete NPC structures (examples, blue arrows). Putative NPC structures (green) markers are automatically identified using the linearized envelope localization histogram (shown in white, Supplementary methods). (**F**) Method for putative NPC structure isolation and alignment. Peaks in the envelope histogram (black line) are identified as potential locations for putative NPC structures (black circles). Localizations falling into a window (width, w, length, l) centred at these locations (p) are cropped out and rotated to a common frame, c, by the angle θ, maintaining the cytoplasm-nucleus vector, n.

The following figure supplement is available for figure 2:

**Figure supplement 1**. Localization precision.

knockdown to study how alterations of Nup153 levels influence the organization of impβ within the pore. siRNA treatment led to ∼70% protein reduction of Nup153 ($\Delta153_{70\%}$, **Figure 3—figure supplement 4**). Indeed, when Nup153 was reduced by siRNA knockdown, the dSTORM signal changed. The impβ map shows fewer impβ localizations overall and marked reduction of signal from the entire nuclear side of the NPC, creating an asymmetric, teardrop-like pattern (**Figure 3A**, 'impβ $\Delta153_{70\%}$'). Therefore, both addition of RanGTP and reduction of Nup153 alter the arrangement and loading of impβ within the pore, especially towards the NPC's nuclear face.

Although population averaging allows major differences to be detected, such averaging can obscure more subtle changes. To compare many NPCs without population averaging, we represented each NPC as a single vertical line whose color corresponds to impβ concentration, ranging from blue to red. We placed each of those lines side-by-side, giving a 'waterfall' plot (**Figure 3E**). The pores with most of the impβ at the nuclear face are at the left of the plots, while the pores with most of the impβ at the cytoplasmic face are to the right. As can be seen, there is considerable pore-to-pore heterogeneity in the axial distribution of impβ in all studied conditions. Some NPCs had a strong impβ signal only at the cytoplasmic face, other NPCs had similar levels of impβ at both faces, and finally, some

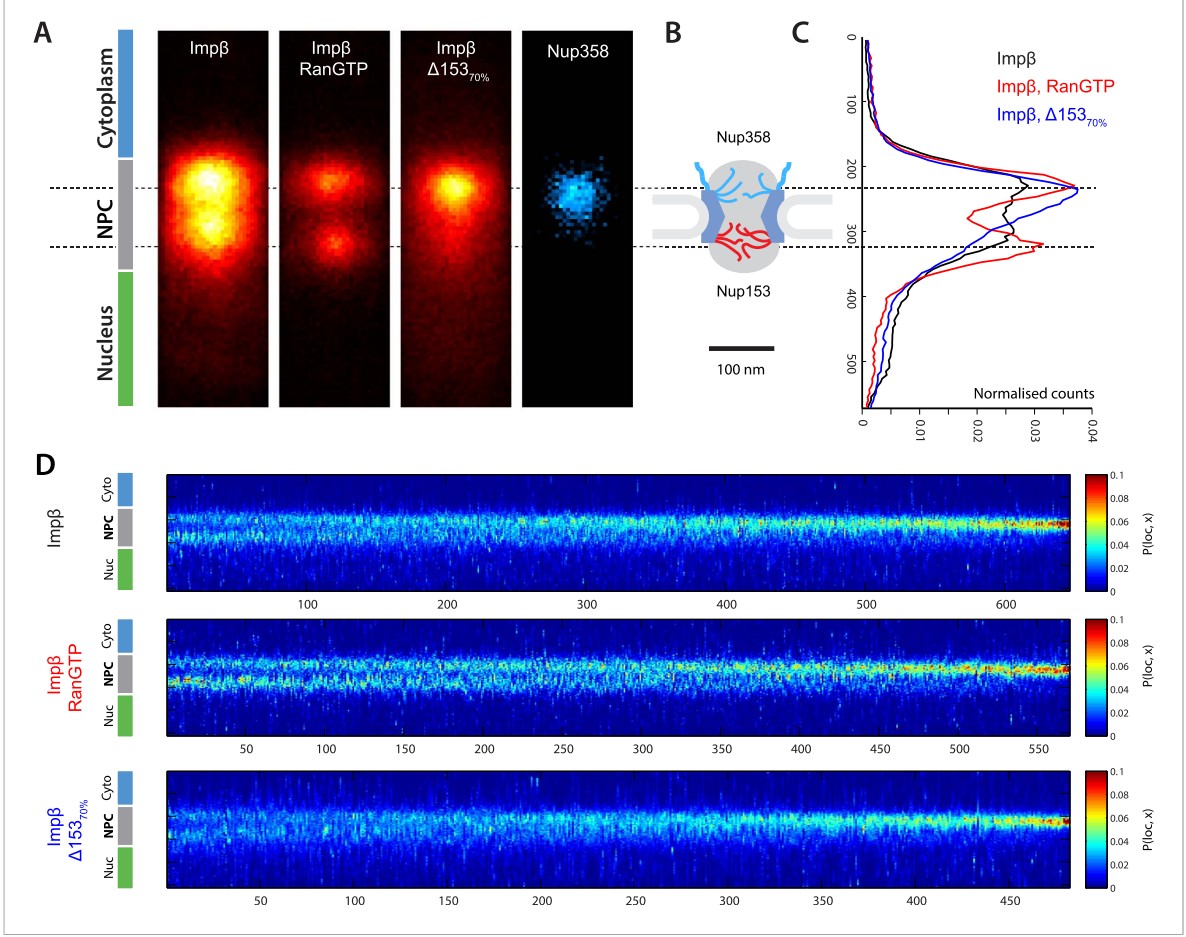

**Figure 3**. Localization microscopy of impβ spatial organisation and its Ran-dependence. (**A**) 2D histograms of impβ density in the NPC under different conditions. To generate these panels, we sum all localizations for a given condition and divide by the number of NPCs per condition. *Figure 3—figure supplement 1* shows examples of the raw localization images for each of the conditions. Anti-Nup358 antibody was localized using a second dye pair, as shown in *Figure 3—figure supplement 2*. Histograms of the number of localizations per NPC, for each condition are shown in *Figure 3—figure supplement 3*. (**B**) Schematic of the NPC showing the tethering locations of Nup358 and Nup153. (**C**) Probability density functions (PDF) of impβ localizations, showing the relative redistribution of impβ localizations in the presence of RanGTP and with Nup153 knockdowns. (**D**) 'Waterfall' plots showing PDF projections of NPC structures in each of the conditions tested. Each column of the plot represents a single NPC structure and is arranged from left to right according to the ratio of cytoplasmic to nuclear localizations.

The following figure supplements are available for figure 3:

**Figure supplement 1**. Examples of raw localization data for each of the conditions.

**Figure supplement 2**. Two color STORM imaging.

**Figure supplement 3**. Histogram of the number of raw localizations per NPC structure.

**Figure supplement 4**. Quantification of siRNA knockdown of Nup153.

NPCs had most of their impβ at their nuclear face. The most notable difference was that when Nup153 was reduced, there were very few pores with a dominant impβ signal at the nuclear face.

Summarizing, in the absence of RanGTP, the pore is loaded with impβ. The addition of RanGTP increases impβ turnover, depleting transport receptors from the pore. Partial knockdown of Nup153 reduces the overall impβ counts and depletes transport receptors from the nuclear side of the pore, resulting in an asymmetric, cytoplasm-biased impβ distribution. Together, these results raise the

possibility that Ran can modulate the interactions between impβ and the NPC. We then turned to a quantitative assessment of impβ levels within the pore, to determine whether Nup153 is a dominant site of RanGTP-sensitive impβ binding.

## Single-molecule photobleach step-counting of impβ at the NPC

Although dSTORM microscopy is able to localize populations of molecules and detect relative changes in their spatial arrangements, it does not allow absolute numbers of molecules to be estimated. We thus used a single-molecule photobleach step-counting assay (*Leake et al., 2006*) to estimate the numbers of impβ molecules displaced by nuclear RanGTP at the NPC. Digitonin-permeabilized nuclei from wild type and $\Delta153_{70\%}$ cells were incubated with impβ-mCherry with or without RanGTP and then fixed, yielding nuclear pores that can be imaged at the basal surface of the nucleus as bright spots (*Figure 4A*). Under the appropriate imaging conditions, discrete single-molecule photobleaching steps can be resolved in the fluorescence bleaching traces of impβ-mCherry at the pores (*Figure 4B*). The photobleaching fluorescence step-size, $x$, for a specific NPC, can be calculated by taking the first peak of the power spectrum of the pairwise difference distribution of the bleaching trace (*Figure 4C*). From the fluorescence step-size and initial intensity, $\Delta I$, of the pore, the relative amount of impβ molecules at a single NPC can be measured. Because of potential systematic errors in determining absolute numbers of impβ molecules with this technique (e. g., homo-FRET, incomplete mCherry maturation), relative analysis of impβ levels was performed by defining the impβ signal of the wild type–RanGTP condition as 100% (*Figure 4D*). Wild type NPCs in the absence of RanGTP contained the greatest number of impβ molecules (70 bleach steps counted), whereas RanGTP caused a 27% decrease in impβ levels (*Figure 4A*, *Figure 4—figure supplement 1*, *Table 1*). In $\Delta153_{70\%}$ nuclei, we found a 34% drop in the amount of impβ per pore without RanGTP and a 33% drop with RanGTP. These results suggest that most of the impβ molecules that are displaced from the NPC by nuclear RanGTP are those that are bound to Nup153.

## Impβ interacts with Nup153 to modulate the NPC permeability barrier and is Ran-sensitive

The detection of a stable RanGTP-sensitive pool of impβ in the pore, the tentative identification of a binding partner, and the quantification of the energy-dependent changes within the pore motivated functional studies seeking to detect possible impβ/RanGTP/Nup153-mediated alterations of passive facilitated diffusion and active transport.

We first investigated Nup153's relevance to impβ-mediated transport using the SA-IBB$_4$ cargo. The cargo was added to digitonin-permeabilized nuclei that either contained impβ only (to monitor passive equilibration of the cargo across the nuclear envelope) or impβ, Ran, NTF2 (the RanGDP importer), and GTP (to monitor active transport). Impβ was added to the nuclei before the cargo, allowing us to examine how cargo molecules translocate through pores that already contain transport receptors. As shown earlier (*Figure 1*), little cargo was able to enter the nucleus under conditions of passive equilibration (i.e., in the absence of RanGTP) in wild-type cells. In contrast, the cargo translocated the NPC faster in $\Delta153_{70\%}$ nuclei, indicating that the transport channel had become leakier to large cargos and translocation became less dependent on the presence of RanGTP (*Figure 5A,B*). Interestingly, the opposite was observed for active transport in the presence of RanGTP, where net nuclear cargo accumulation was reduced for $\Delta153_{70\%}$ nuclei (*Figure 5A,B*). The nucleoporin Nup153 therefore affects both the ease of passive impβ-mediated movement of large cargos through the pore and the efficiency of active transport into the nucleus.

To further characterize the permeability barrier within the NPC, we determined whether impβ and RanGTP affect the free diffusion of cargos through the pore under both normal and reduced levels of Nup153. We employed a series of inert probes consisting of single GFPs, GFP dimers, and GFP trimers (GFP$_1$, GFP$_2$, and GFP$_3$) with molecular masses of 27, 54, and 83 kDa respectively. Because these probes do not contain an IBB and cannot bind impβ, they exclusively undergo passive transport.

We found that 1 μM impβ significantly decreased the permeability of the NPC for the GFP$_1$ and GFP$_2$ probes (*Figure 5C,D*). By contrast, when RanGTP was also added, the permeability was greatly increased. The GFP$_3$ probe translocated across the NPC at a relatively slow rate with or without impβ and RanGTP, likely because GFP$_3$'s size is considerably larger than the passive diffusion size cutoff of the pore (*Figure 5C*). We therefore decided to focus on the GFP$_2$ probe and we used it to further explore the effects of impβ and Nup153 on the permeability of the NPC (*Figure 5D,E*). First, we

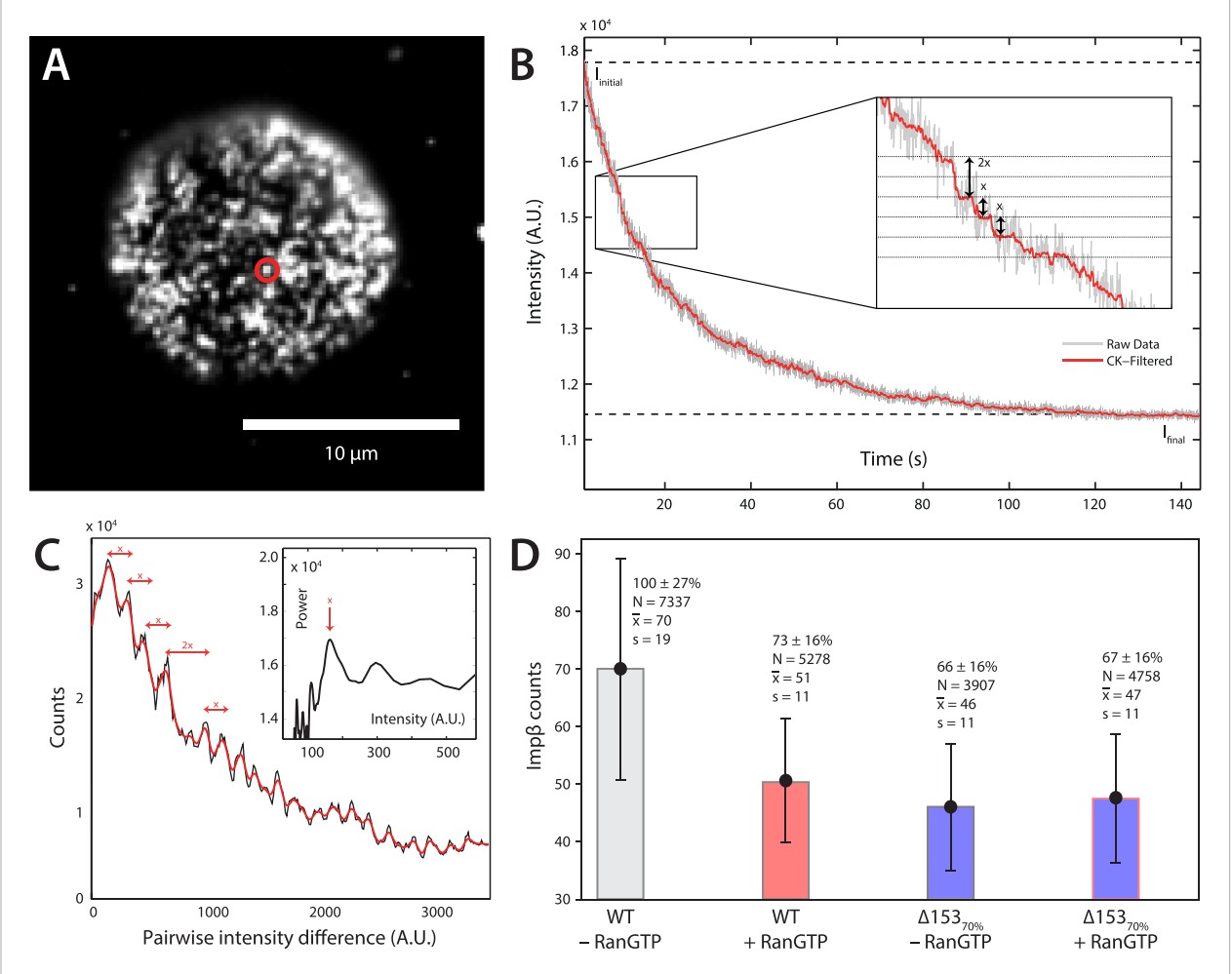

**Figure 4**. Photobleach step-counting of impβ at the NPC. (**A**) A 200-frame average image of impβ-mCherry at the basal envelope of the nucleus. Individual NPCs can be identified (example highlighted in red). (**B**) Fluorescence intensity vs time trace for the NPC highlighted in (**A**) under continuous illumination. The raw intensity signal is shown in gray and the Cheung-Kennedy filtered signal is shown in red. Inset: individual photobleaching steps (x) can be clearly identified. (**C**) Pairwise difference distribution function calculated from the intensity trace shown in (**B**). Characteristic step sizes can be identified from the peaks in the distribution. Inset: the power spectrum of the pairwise difference distribution, showing the characteristic intensity signal for a single mCherry, x. We can then calculate the number of molecules by dividing the total intensity change in the trace in (**B**) by the fluorescence intensity of a single molecule calculated from (**C**). (**D**) Impβ counts as a function of RanGTP and Nup153. Addition of RanGTP and reduction of Nup153 decrease the impβ counts by approximately equal amounts. Normalized impβ abundance, number of pores analyzed, mean, and standard deviations are denoted. Error bars represent standard deviations about the means.

The following figure supplement is available for figure 4:

**Figure supplement 1**. Distribution of count values.

tested the possibility that general molecular crowding, for instance due to widespread transport receptor-FG nucleoporin interactions, was responsible for the observed permeability modulation by impβ. We therefore repeated the previous experiments with the related transport receptor transportin-1 (TRN1). TRN1 is the transport receptor for M9 signal peptide-containing cargos such as hnRNPs and belongs to the same class of karyopherins as impβ (*Pollard et al., 1996*). At 1 and even 2 μM TRN1, there was no strong effect on NPC permeability (*Figure 5E*), suggesting that general molecular crowding is not responsible for the changes to NPC permeability. Importantly, for $\Delta153_{70\%}$ nuclei, addition of impβ no longer restricted the NPC (*Figure 5D*). Together, these results suggest that specifically the impβ•Nup153 interaction causes the nuclear pore to become less permeable.

**Table 1**. Mann–Whitney confidence intervals of impβ changes seen in ±RanGTP and ±Δ153$_{70\%}$ conditions

| | WT − RanGTP | WT + RanGTP | Δ153$_{70\%}$ − RanGTP | Δ153$_{70\%}$ + RanGTP |
|---|---|---|---|---|
| WT − RanGTP | – | [17.8, 18.9] | [22.5, 23.7] | [21.1, 22.2] |
| WT + RanGTP | [−17.8, −18.9] | – | [4.5, 5.4] | [2.9, 3.8] |
| Δ153$_{70\%}$ − RanGTP | [−22.5, −23.7] | [−4.5, −5.4] | – | [−0.9, −1.8] |
| Δ153$_{70\%}$ + RanGTP | [−21.1, −22.2] | [−2.9, −3.8] | [0.9, 1.8] | – |

For example, the addition of RanGTP compared to baseline results in a *drop* of about 18 impβ counts (column 1, row 2); The Mann–Whitney confidence interval is [−17.8, −18.9].

Furthermore, because the inert probes undergo purely passive translocation across the NPC, the reduced permeability must be due to a specific steric 'barrier' within the pore and not due to a block of transport receptor-specific binding sites. This steric barrier appears to involve impβ•Nup153 interactions that are very stable and long-lived in the absence of RanGTP. Indeed RanGTP causes the pore to become more permeable even when no exogenous impβ is first added. This is likely due to endogenous impβ and other transport receptors residing in the pore that were not washed away during digitonin permeabilization (*Figure 5—figure supplement 1*).

Along with the observation that impβ can persist in the pore for minutes or tens of minutes, these functional studies suggest that impβ could be considered a *bona fide* functional component of the pore and not only a soluble transport receptor. Moreover, the impβ•Nup153 interaction may be responsible for the permeability differences detected in our inert probe passive diffusion assays and may contribute to the permeability barrier function of the NPC.

## Impβ and Nup153 form higher-order complexes in vitro that are dissolved by RanGTP

To explore the notion that impβ and Nup153 act together to form a Ran-sensitive permeability barrier, we investigated their interaction in vitro. Upon co-incubation of recombinant impβ and Nup153FG (the FG domain of Nup153 comprising amino acids 874–1475 [*Lim et al., 2006*]), large, micron-sized structures formed on a timescale of minutes (*Figure 6A*). We turned to fluorescence fluctuation spectroscopy (*Chen et al., 2000*; *Tetin, 2013*) to examine the structure's assembly and disassembly behaviors and requirements. The fluorescence intensity signal of diffusing impβ-YFP molecules (50 nM) showed a fluctuation pattern characteristic of freely diffusing proteins (*Figure 6B*). However, when Nup153FG (0.5 µM) was added, large intensity bursts appeared within tens of seconds. The appearance of these spikes in intensity (along with their corresponding long tails in the photon counting histograms) indicated the formation of large impβ•Nup153 complexes (*Figure 6B,C*, red traces). These higher-order complexes were orders of magnitude brighter than the freely diffusing impβ-YFP, suggesting that they are comprised of tens or even hundreds of impβ molecules. The formation of large complexes can be explained by the many FG motifs found in Nup153's FG domain as well as the multiple sites on impβ's surface that may bind FG repeats. Notably, the addition of RanQ69L•GTP (2 µM), which does not hydrolyze GTP (*Bischoff et al., 1994*) and is therefore stably in the GTP-bound form, entirely inhibited formation of the complexes. RanQ69L•GTP even dissolved existing large impβ•Nup153FG complexes (*Figure 6—figure supplement 1A*). This Ran action occurred specifically through impβ (and not Nup153FG) binding since Nup153FG in complex with an impβ truncation lacking the Ran-binding domain, impβ(ΔN70), became insensitive to RanQ69L•GTP (*Figure 6—figure supplement 1B*).

Performing similar experiments with TRN1-GFP (100 nM), we again detected spikes in intensity upon addition of Nup153FG, indicating the formation of large complexes (*Figure 6D*). This is not surprising given that TRN1 is structurally similar to impβ and likely also contains multiple FG-binding sites (*Chook and Blobel, 1999*). However, although TRN1 and RanGTP are known binding partners (*Chook and Blobel, 1999*), RanQ69L•GTP had no observable effect on the TRN1•Nup153FG complexes, suggesting a functional difference between TRN1 and impβ. Indeed, it has been previously reported that TRN1-mediated nuclear import is less dependent upon the RanGTP gradient

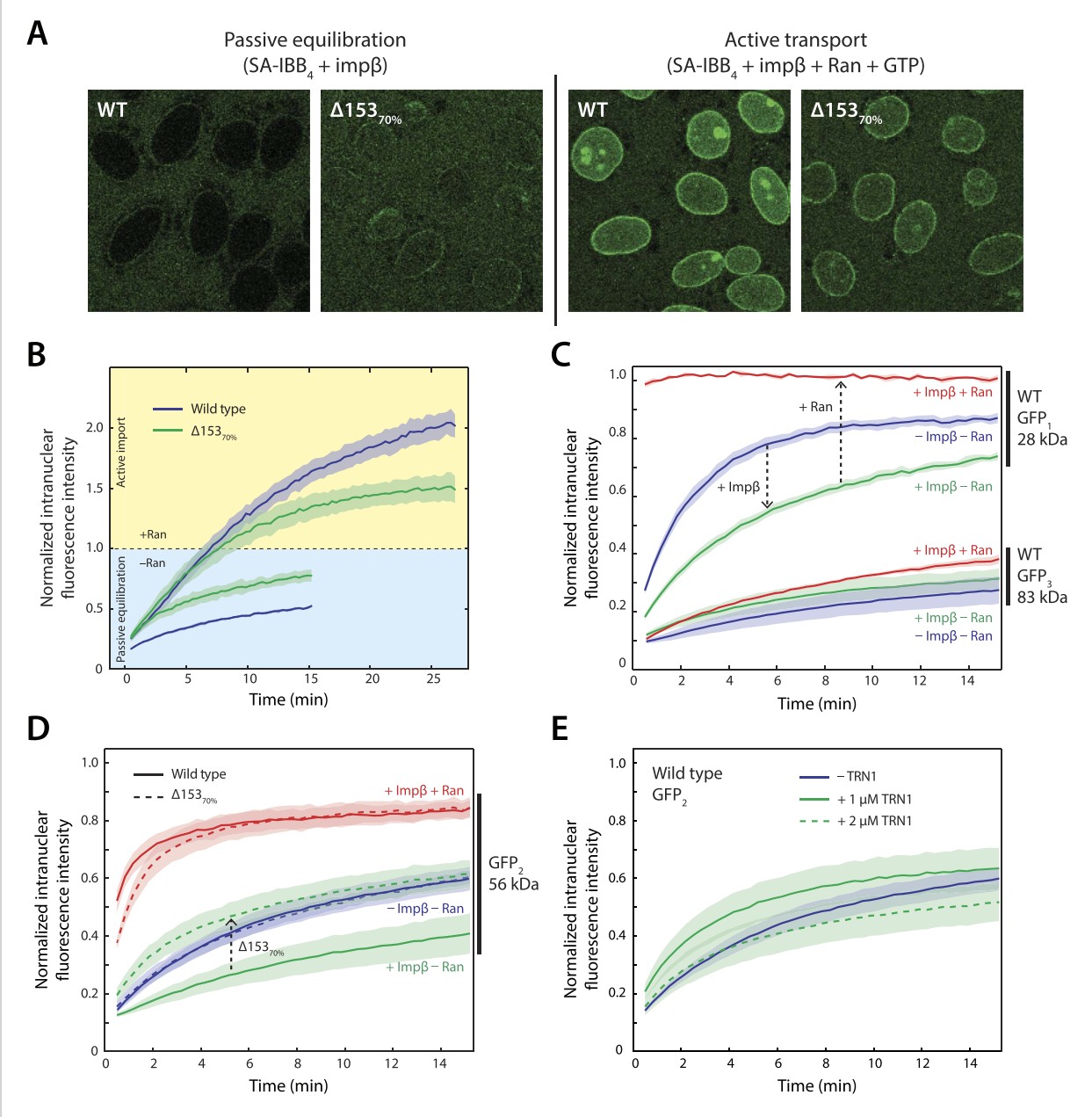

**Figure 5**. Effect of Nup153 reduction on active transport and passive equilibration. (**A**) Confocal fluorescence microscopy images showing the change in distribution of a fluorescently labeled 220 kDa SA-IBB$_4$ cargo as a function of Nup153 knockdown. (**B**) Reduction of Nup153 enables the cargo to passively equilibrate (pale blue region) more rapidly through NPCs loaded with impβ. However, reduction of Nup153 impairs the active transport of SA-IBB$_4$ (pale yellow region). (**C**) Passive equilibration of the inert GFP$_1$ and GFP$_3$ probes as a function of impβ (1 μM) and RanGTP (5 μM). For GFP$_1$, note the rate decrease by impβ and the rate increase with RanGTP. For GFP$_3$, passive equilibration is slow in all conditions. (**D**) Passive equilibration of GFP$_2$ as a function of impβ, RanGTP, and Nup153. For wild-type cells, impβ and RanGTP have similar effects on GFP$_2$ as on GFP$_1$. For Δ153$_{70\%}$ cells, impβ no longer slows passive diffusion of GFP$_2$. RanGTP, however, still facilitates equilibration. (**E**) TRN1 does not significantly slow GFP$_2$ diffusion through the NPC at 1 or even 2 μM. In all plots, shaded regions indicate the standard error of the mean (N ≥ 3 for all conditions).

The following figure supplement is available for figure 5:

**Figure supplement 1**. RanGTP and a Ran 'wash' increase the passive equilibration of GFP$_2$ into wild type nuclei even when no impβ is present.

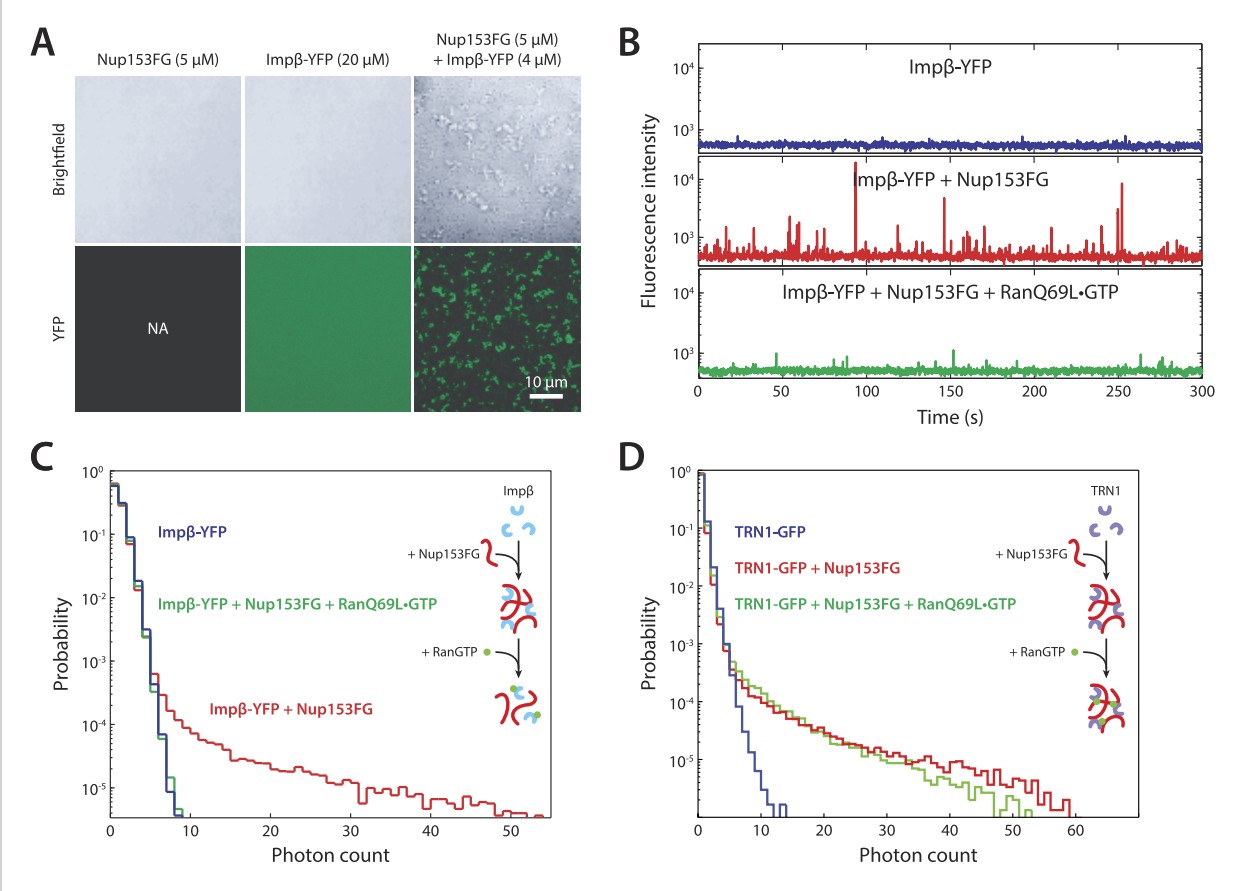

**Figure 6**. In vitro formation of large RanGTP-reversible impβ•Nup153FG complexes. (**A**) Confocal images of impβ•Nup153FG complexes. Brightfield and YFP fluorescence images of Nup153FG (left), impβ-YFP (center), and Nup153FG + impβ-YFP (right). Complexes form only when both proteins are co-incubated. (**B**) Fluorescence fluctuation intensity traces of impβ-YFP in the presence of Nup153FG and RanQ69L•GTP. Large intensity bursts appear with the addition of Nup153FG but are inhibited in the presence of RanQ69L•GTP. (**C**) Photon counting histograms for the experiments shown in (**B**). Impβ and Nup153FG form large, bright complexes (note the extended tail with Nup153FG); however, these complexes disappear when RanQ69L•GTP is added (see inset schematic). (**D**) Photon counting histograms of fluorescence fluctuations for TRN1-GFP in the presence of Nup153FG and RanQ69L•GTP. Large complexes also form between TRN1 and Nup153FG but are not affected by RanQ69L•GTP (see inset schematic).

The following figure supplement is available for figure 6:

**Figure supplement 1**. Additional fluctuation traces.

than impβ-mediated import (*Ribbeck et al., 1999*). Moreover, the passive diffusion assays (*Figure 5E*) did not detect alterations of pore permeability in the presence of TRN1, suggesting that TRN1 either does not form a meshwork inside the pore or that a hypothetical TRN1-mediated barrier has significantly different biophysical characteristics (e.g., effective pore size) compared to the one formed by impβ. In support of both those possibilities, TRN1 has been shown to bind different sites within the Nup153 FG domain relative to impβ (*Shah and Forbes, 1998*). Therefore, despite the fact that impβ and TRN1 are both transport receptors and that both can bind Nup153 and form higher-order complexes with it, previous reports (*Shah and Forbes, 1998*; *Ribbeck et al., 1999*) and the data shown in *Figures 5E*, *6D* point to TRN1 being functionally and biophysically distinct from impβ.

## Discussion

The importance of members of the impβ family of transport receptors and Ran in active nuclear transport has been firmly established for many years. The basic import reaction involves the RanGTP-driven displacement of impβ family members from their cargo in the correct compartment.

However, there are multiple reports that RanGTP and impβ may have critical additional roles in passive equilibration and active transport. Specifically, it was proposed that Ran is necessary for impβ-bearing cargos to move past a barrier located at 70 nm along the transport axis (*Gorlich et al., 1996*; *Lowe et al., 2010*). Based on the experiments reported here, we propose the existence of a Ran-sensitive network of interactions between impβ and Nup153 centered at the nuclear face and including the central channel of the NPC, which contributes to the permeability of the pore (*Figure 7*). Whilst other nuclear basket localized Nups, such as Nup50, have been shown to promote cargo dissociation from the pore in active transport (*Sun et al., 2008*), Nup153 appears to also have a role in controlling bulk permeability of the NPC. The impβ•Nup153 interaction significantly restricts the ability of inert cargos to diffuse across the NPC, indicating the presence of a non-specific physical barrier that cannot be explained by simple molecular crowding. This barrier may take the form of a highly cross-linked 'meshwork' of long, flexible Nup153 FG domains fastened to each other by impβ molecules, which we characterized in vitro at physiological pH and salt concentrations. The multiply cross-linked nature of these impβ•Nup153 structures may be reminiscent of the FG gel materials reported by others (*Frey et al., 2006*; *Schmidt and Gorlich, 2015*). However, these FG gels are held together by homotypic interactions between the FG domains; here, materials form via specific coordination between impβ and the FG domains. Moreover, the resulting impβ•Nup153 material is dynamic, in the sense that its formation and final stability is sensitive to RanGTP, which can even dissolve existing large impβ•Nup153 complexes (*Figure 6—figure supplement 1*). The in vitro fluorescence fluctuation data obtained through spectroscopic studies of purified proteins correlate well with our localization microscopy studies of impβ's spatial distribution within the pore, where we see a sub-population of impβ in the channel that is significantly reorganized by RanGTP. Furthermore, the photobleach counting experiments with wild type and reduced Nup153 NPCs suggest that this Ran-sensitive pool is predominantly located at Nup153, although the counting experiments do not rule out other RanGTP-sensitive impβ binding sites within the pore. The sub-second turnover kinetics of the RanGTP-sensitive impβ pool (*Figure 1*) are similar to kinetic values for impβ turnover inside living cells (*Rabut et al., 2004*), suggesting that our reconstituted 'in vitro' permeabilized cell transport assay recapitulates key features of transport in intact living cells.

Two quantitative imaging approaches, a two-color photo-conversion approach and single-protein counting, were used to investigate an extremely stable subpopulation of impβ which cannot be easily detected by other methods due to the reaction timescale, photo-bleaching effects, and limitations of instrument stability. We estimate that on average, 73 ± 16% of impβ molecules in each pore are insensitive to nuclear RanGTP and bound stably to the NPC for many minutes. These results, coupled with our observation that impβ contributes to the NPC's permeability, suggest that impβ is a functional component of the pore and does not just facilitate cargo translocation. Indeed, other theoretical and experimental studies have suggested that transport receptor binding at the NPC plays

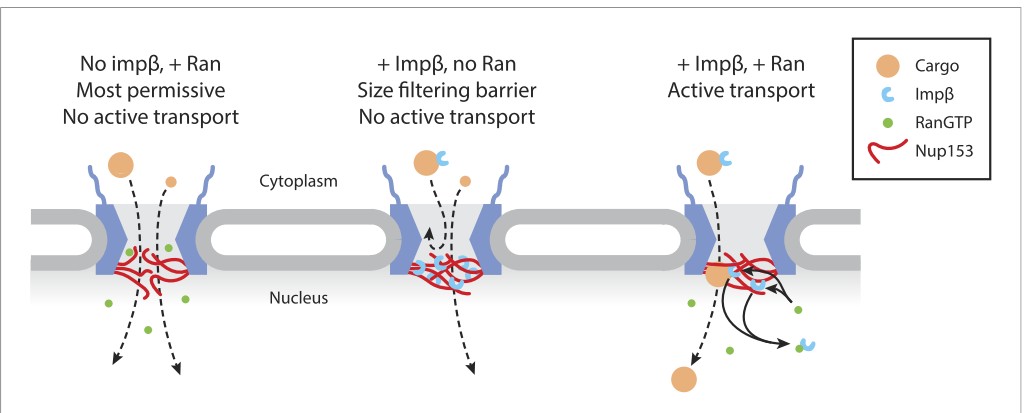

**Figure 7**. Model of Ran-sensitive impβ•Nup153 interactions at the nuclear face of the NPC. In this model, multivalent interaction of impβ with Nup153 yields a cross-linked mesh that restricts the movement of inert molecules and cargo-receptor complexes. This impβ•Nup153 barrier is modulated by Ran.

a critical role in the non-specific occlusion of inert cargos from entering the pore (*Zilman et al., 2007*; *Jovanovic-Talisman et al., 2009*; *Zilman and Bel, 2010*; *Schleicher et al., 2014*, *Kapinos et al., 2014*). A functional role of impβ within the NPC is particularly intriguing in light of the structural and functional relationship between karyopherins and scaffold nucleoporins (*Andersen et al., 2013*; *Sampathkumar et al., 2013*; *Stuwe et al., 2014*), suggesting that these two classes of proteins may share a common evolutionary ancestry. The possible functional and structural roles of the Ran-insensitive impβ pool remain to be discovered. At present, we only know (1) that the bulk of these Ran-insensitive impβ molecules are located near the cytoplasmic face of the pore and (2) that they are not bound to Nup153, since reduction of Nup153 (i.e., $\Delta 153_{70\%}$) did not reduce the impβ counts relative to the RanGTP condition.

Although we have emphasized the 'average' characteristics of the NPC vis-à-vis changes in RanGTP levels and other experimental manipulations, the single-pore resolution dSTORM and counting experiments revealed significant NPC-to-NPC variation. The distributions for our counting experiments give an idea of the heterogeneity of impβ binding amongst NPCs even within one nuclear envelope. The variability of NPC composition may reflect variation of the instantaneous functional state of the cellular pool of several hundred NPCs; perhaps, not all NPCs are functionally equivalent at all times. In the future, it will be interesting to directly relate NPC-to-NPC variation of molecular composition to possible variation of functional transport characteristics.

The observation that the permeability of the NPC, specifically its size-filtering, is sensitive to RanGTP levels is interesting in light of recent results that suggest that cells might actively regulate the RanGTP gradient. The RanGTP gradient is generated and maintained by the chromatin-associated RCC1 exchange factor (*Bischoff and Ponstingl, 1991*) whose activity as well as the local concentration of its RanGDP substrate is subject to multi-tiered regulation (*Li and Zheng, 2004*; *Hood and Clarke, 2007*; *Yoon et al., 2008*; *Hitakomate et al., 2010*). Overall, our studies raise the possibility that the cell might have an extra layer of control over nucleocytoplasmic transport processes by regulating the spatiotemporal characteristics of the RanGTP gradient, which would then modulate both the size-cutoff of passive permeability and the extent of active transport.

Beyond clarifying the RanGTP-dependent composition and organization of the intact pore, the ability to form the Ran-reversible impβ/Nup153 material in vitro will allow the interplay of energy, impβ, and Ran to be directly investigated and should also allow efficient and selective molecular rectifiers to be created in vitro, not just for biological cargos but also for other substrates.

# Materials and methods

## Plasmids; protein expression and purification

### Plasmid construction

Plasmids were synthesized using the SLIC procedure (*Li and Elledge, 2007*). DNA primers were purchased from Elim Biopharmeuticals, Inc. XL1-Blue chemically competent *Escherichia coli* cells were transformed and selected for by antibiotic resistance. Plasmids were purified using the Qiagen QIAprep Spin Miniprep Kit and sequenced. Constructs and Plasmids are listed in *Supplementary file 1*.

### Protein expression and purification

Proteins were expressed and purified as detailed previously (*Lowe et al., 2010*). Briefly, *E. coli* (BL21 DE3) were transformed with the appropriate plasmid and grown in 1 l of LB media with the appropriate antibiotic. The cells were grown at 37˚C to an A600 of ∼0.6 and then cooled to room temperature. Protein expression was induced with 0.5 mM IPTG overnight. Biotinylated proteins were expressed in the presence of 0.1 mM biotin and a biotin ligase. Cells were harvested by centrifugation at 5000×*g* at 4˚C for 15 min, and the pellet was resuspended in PBS (pH 7.4) containing 20 mM imidazole, 1 mM β-mercaptoethanol, and protease inhibitors (Complete Protease Inhibitor Cocktail Tablet, Roche Diagnostics Corporation, Indianapolis, IN). Proteins were purified by Ni-NTA affinity chromatography, followed by size-exclusion chromatography (Superdex 75, GE Healthcare, Pittsburgh, PA). Proteins were typically dialysed into XB buffer (10 mM HEPES pH 7.7, 1 mM $MgCl_2$, 100 mM KCl, 50 mM sucrose), flash frozen in liquid nitrogen, and stored at −80˚C. Protein purity was judged by SDS-PAGE, and concentrations determined by UV absorbance (using calculated extinction coefficients) or Bradford assays. Nucleotide loading of Ran was performed as described previously (*Askjaer et al., 1999*). Briefly, Ran was incubated for 40 min on ice with 6 mM EDTA and a 50-fold

excess of nucleotide (GDP or GTP). The reaction was stopped with a final concentration of 25 mM $MgCl_2$ added slowly (in four portions in 1 min intervals). The protein was then dialysed against 30 mM potassium phosphate pH 7.6, 2 mM Mg-acetate, 2 mM GDP or GTP, 7% glycerol, and 2 mM β-mercaptoethanol, at 4°C overnight. For Nup153FG purification, the cell lysate was run over a 5-ml GSTrap HP column (GE Healthcare) equilibrated in PBS. Bound protein was eluted with 10 mM reduced glutathione in 50 mM TrisHCl pH 8.0. The sample was concentrated and loaded onto a HiPrep 16/60 Sephacryl S-300 High Resolution size exclusion column (GE Healthcare) equilibrated in 25 mM HEPES pH 7.5, 400 mM NaCl, 10% glycerol, 1 mM DTT, flash frozen, and stored at −80°C.

## Labeling, imaging buffers, cell culture, and import assays

### Protein and antibody fluorescent dye labeling

Purified proteins were labeled for dSTORM using N-hydroxysuccinimidyl esters of Alexa649, Cy5, or additionally with Alexa 405/488/532 for multicolor STORM, according to the manufacturers' protocols. Antibodies were purchased from Abcam (Cambridge, UK): Anti-Nup153 antibody [SA1] [ab96462], Anti-RanBP2 antibody [ab64276], Donkey polyclonal Secondary antibody to Rabbit IgG—H&L [ab6701], Donkey polyclonal Secondary antibody to Mouse IgG—H&L [ab6707].

### Antibody labeling protocol

HeLa cells were washed three times with PBS and then fixed in 4% PFA for 15 min. The PFA was removed and the cells were washed 3 × 2 min in PBS with agitation (70 RPM on a rotary shaker). Cells were permeabilized with 0.5% Triton X-100 for 5 min at RT followed by 3 × 2 min PBS washes. The cells were incubated in blocking buffer (PBS + 10% vol/vol goat/donkey serum + 1.25 mg/ml BSA) for 1 hr at RT. The antibodies were diluted according to manufacturers' suggestions in blocking buffer. The cells were incubated with the primary antibody for 30 min, washed 3 × 5 min with blocking buffer, incubated with the secondary antibody for 30 min, and washed 3 × 5 min with PBS.

### STORM and dSTORM imaging buffers

Imaging was performed using the following buffer conditions: 10/100 mM mercaptoethylamine (Sigma–Aldrich), 0.5 mg/ml glucose oxidase (Sigma–Aldrich, St. Louis, MO), 0.2% vol/vol catalase (Sigma–Aldrich), 10% wt/vol D-Glucose in PBS pH 7.4.

### Cell culture

HeLa cells were cultured in DMEM media supplemented with 10% FBS. Cells were plated on glass-bottomed (size 0 thickness) poly-lysine-coated chambers (MatTek Corporation, Ashland, MA) at a seeding concentration of $2.5 \times 10^5$ cells/ml the day prior to use.

### Import assays

Import assays were performed as reported previously (*Lowe et al., 2010*). The buffers used were PBS (137 mM NaCl, 2.7 mM KCl, 8 mM $Na_2HPO_4$, 2 mM $KH_2PO_4$, pH 7.4), permeabilization buffer (50 mM HEPES, 50 mM KOAc, 8 mM $MgCl_2$, pH 7.3), and transport buffer (20 mM HEPES, 110 mM KOAc, 5 mM NaOAc, 2 mM MgOAc, 2 mM DTT, pH 7.3). The cell permeabilization protocol is based on that of *Adam et al. (1990)*. The cells were washed for 3 × 2 min with PBS, followed by a 2-min wash with permeabilization buffer, followed by a 5-min permeabilization with digitonin (Sigma–Aldrich) at a concentration of 50 μg/ml supplemented with an energy regenerating system of 100 μM ATP (Roche), 100 μM GTP (Roche), 4 mM creatine phosphate (Roche), and 20 U/ml creatine kinase (Roche) in permeabilization buffer. The digitonin was subsequently removed by washing for 3 × 3 min with transport buffer. After the final wash, excess liquid was removed and the appropriate experimental reaction mix was quickly added to the nuclei. Control experiments with fluorescently (FITC) labeled dextrans (70 kDa) were used to confirm that the nuclear envelope remained intact following the digitonin permeabilization.

### Active nuclear import assays

Digitonin-permeabilized HeLa cells were treated with an active nuclear import reaction mix containing a fluorescent import cargo probe, importin-β (various concentrations), RanGDP (5 μM), NTF2 (4 μM), and an energy regenerating system (2 mM GTP, 0.1 mM ATP, 4 mM creatine phosphate, and 20 μ/ml creatine kinase) in transport buffer. Import reactions proceeded at room temperature for 20 min before the cells were fixed with a 4% PFA solution for 15 min and washed 3 × 2 min with PBS. Cells were then imaged using a Zeiss LSM 700 confocal laser scanning microscope (Carl Zeiss AG, Oberkochen, Germany).

## Passive nuclear import assays

Digitonin-permeabilized HeLa cells were treated with a passive nuclear import reaction mix containing a fluorescent passive import probe (either 1xGFP, 2xGFP, or 3xGFP) and also (depending on the experimental condition) importin-ß (1 µM), RanGDP (5 µM), NTF2 (4 µM), and an energy regenerating system (2 mM GTP, 0.1 mM ATP, 4 mM creatine phosphate, and 20 µ/ml creatine kinase) in transport buffer. Passive import reactions were imaged live for 15 min (at 20 s intervals) using a Zeiss 700 LSM laser scanning confocal microscope.

## RNA knockdown

RNA interference was used to knock down protein expression of Nup153 using an siRNA corresponding to nucleotides 2593–2615 of human Nup153 (5′-AAGGCAGACUCUACCAAAU GUdTdT-3′) (*Harborth et al., 2001*; *Zhou and Pante, 2010*). HeLa cells were plated in glass-bottom dishes (MatTek) at a density of $1.25 \times 10^5$ cells/dish (2 ml volume) the day prior to siRNA transfection. Lipofectamine RNAiMAX Transfection Reagent (Invitrogen, Carlsbad, CA) was used following the manufacturer's protocol. Briefly, 5 µl of Lipofectamine reagent was diluted 50-fold into Opti-MEM I Reduced Serum Media (Invitrogen). 75 pmol of siRNA was diluted into an equal volume of Opti-MEM media. The Lipofectamine and siRNA were then mixed together, incubated at room temperature for 10 min, and then added to the cells. Cells were used for experiments ~48 hr after transfection as no noticeable difference was observed past 48 hr. Knockdown efficiency was determined to be about 70% using immunofluorescence (measured to be 68%) and Western blot (measured to be 72%) (*Figure 3—figure supplement 2*).

# General imaging hardware and analysis

## Confocal imaging, photoconversion, and bleaching

Imaging was performed on a Zeiss 700 confocal laser scanning microscope. GFP constructs were imaged using the 488 nm laser. Photo-convertible mEos2 constructs were imaged in two separate channels, with localized photoactivation performed using the 405 nm laser.

## Confocal time-series analysis

Image analysis was performed using custom-written MATLAB (The MathWorks Inc., Natick, MA) scripts for quantifying fluorescence intensities. Briefly, a mean value of intranuclear fluorescence intensity was calculated for each nucleus in the image using an automated nucleus segmentation algorithm. For passive import assays, the nuclear fluorescence intensity value was normalized against the background fluorescence intensity.

## Photobleaching hardware

FRAP was performed on a custom built microscope (*Figure 1—figure supplement 2*). The microscope allows one to perform simultaneous high-speed widefield imaging with a controlled diffraction limited bleaching/photoconversion spot at the center of the field of view. Briefly, four lasers (100 mW 405 nm Coherent Cube, and 100 mW 488, 514 and 561 nm Coherent Sapphires, Coherent Inc., Santa Clara, CA) were combined and expanded to a similar beam diameter. Each laser was under the controller of a shutter. Half-wave plates allow for adjustment of polarization. A polarizing beam splitting cube (PBS) splits the beam into two 'arms'. The 'focused spot' arm passes via a matched pair of convex lenses (f = 50 mm), one of which is mounted on a Z-translation stage to modify the focal depth position. An additional shutter in this arm allows for control of timing of the activation. The 'imaging' arm has an additional lens (f = 200 mm) in order to focus the beam at the back focal plane of the objective (BFP). The two paths are recombined using a second PBS and pass through a quarter wave plate before the remainder of the TIRF lens system to the objective (Olympus 60× 1.49 N.A. TIRF apochromatic objective, Olympus Corporation, Tokyo, Japan) via a multi-edge dichroic filter (Semrock Inc., Rochester, NY). An actively cooled EMCCD camera (iXon+ or iXon Ultra, Andor Technology, Belfast, UK) was coupled to the camera port of the microscope via an additional magnifier. Sample positioning was controlled via a motorized stage with an additional XYZ-Nanopositioning stage (Physik Instrumente, Karlsruhe, Germany) for fine control. All software to control the microscope was written in C++.

## FRAP analysis

Images were processed using custom written MATLAB code, which automated identification of the cell and of the nuclear envelope. The image processing code generates masks for the bleached and

unbleached portions of the nuclear envelope as well as the background. These masks were used to extract raw intensity traces for the three regions. Raw intensity traces for the photobleached region of the nuclear envelope were normalized using the unbleached portion of the nuclear envelope to correct for 'background' bleaching caused by the imaging laser. We then scaled the recovery curve such that the initial pre-photobleach value is 1 and the value immediately after the photobleaching pulse zero. Normalized recovery traces were then used to compute mean recovery trace for each experimental condition.

### Photoconversion analysis

Images were processed using a custom-written MATLAB script that locates the cell in the field of view and creates a mask for the region corresponding to the nuclear envelope and the photoconverted region of the envelope. The mask for the photoconverted region of the nuclear envelope was used to compute the mean intensity in this region in both red and green channels at each time point. The red channel intensity of the region of interest prior to photoconversion (frames 1–4) was averaged to determine the background, which was typically undetectably low. The 405 nm photoconversion laser pulse increases the red intensity, which then decays as the photoconverted molecules leave the pore. The red and green traces provide quantitative information about the kinetics of imp-β turnover in the pore.

## Super-resolution hardware and algorithms

### STORM/dSTORM imaging

All super-resolution imaging was performed on a custom built microscope, based on a Nikon TE-2000 base. Three lasers (100 mW 488 nm Coherent Sapphire, 100 mW 532 nm Coherent Compass, and 100 mW 640 nm Coherent Cube), each with their own shutter control, were expanded to the same diameter and combined using a series of dichroic mirrors into a single free-space beam. Half-wave plates were used to adjust the polarization before passing the beams through an Acousto-Optical Tunable Filter (AOTF, AA Optoelectronics, France) to quickly modulate laser power. The combined beams were again expanded and passed through a quarter-wave plate to circularly polarize the beam. For two-color STORM imaging experiments, we added an additional 405 nm laser (100 mW Coherent Cube), via an optical fiber. The free beam then passed through the TIRF lenses and was focused directly onto the back focal plane of the objective (Olympus 60× or 100× 1.49 N.A. TIRF apochromatic objective) via a multi-edge dichroic filter (Semrock). We used the HILO method of illumination (*Tokunaga et al., 2008*) to image a thin plane through the nucleus. An actively cooled EMCCD camera (iXon+ or iXon Ultra) was coupled to the camera port of the microscope via an additional magnifier. Laser shutter, AOTF, and camera firing were synchronized using a Data Translation DT9834 data acquisition module. Sample positioning was controlled via a micrometer stage with a XY-Nanopositioning stage (Mad City Labs Inc., Madison, WI or Physik Instrumente). Focal drift during image acquisition was corrected using an Objective Z-Nanopositioning stage (Mad City Labs or Physik Instrumente). Camera acquisition was at 40–120 Hz. All software to control the microscope was written in C++ and Python. Data analysis was performed in MATLAB, C++, or Python. The source code for the microscope control software is available at https://github.com/jliphard/OctopusScopeControl.git and other materials (such as MATLAB scripts) are available at http://liphardtlab.stanford.edu/materials.html and at https://github.com/quantumjot/.

### Sub-pixel localization of single-molecules

For a sub-wavelength diameter fluorescent molecule, fitting of the point spread function (PSF) to a Gaussian function yields the highest accuracy and precision of localization (*Cheezum et al., 2001*). Each PSF in successive STORM/dSTORM movie frames was fitted to a symmetrical 2D Gaussian function:

$$f(x,y) \approx A e^{-\left(\frac{(x-x_0)^2}{2\sigma_x^2} + \frac{(y-y_0)^2}{2\sigma_y^2}\right)} + B,$$

where $A$ is the amplitude, $B$ is the background, $x_0$ and $y_0$ are the mean $x$ and $y$ positions, and $\sigma_x$ and $\sigma_y$ are the standard deviations in $x$ and $y$ (where $x = y$ for symmetrical Gaussian functions).

### Drift correction

Drift correction was split into two parts: (i) Real-time focus locking performed during the imaging and (ii) post-imaging translational drift correction:

## Real-time focus lock

Fluorescent beads (0.2 μm Yellow-Green FluoSpheres, Invitrogen) were immobilized to the glass surface of the chamber. The relative z-displacement of the equatorial imaging plane of the nucleus to the surface beads was measured. Imaging proceeded by alternating between imaging the surface beads and correcting for focus drift at the sample surface, and moving up to the imaging plane and performing dSTORM/STORM imaging. Typically, focus drift was stabilized during the experiment, to within ~50 nm using this method.

## XYZ stage translational drift correction

By tracking the fiducial markers at the sample surface plane over time, we can filter and interpolate their trajectories in order to correct the imaging plane movie sequences. We used the interpolated mean fiducial position in order to perform a per-frame drift correction. Typically, translational drift over the experiment was stabilized to <5 nm.

## Localization precision

The localization precision of our instrument refers to how precisely we can define the center of the PSF, given the magnification and signal to noise ratio of the image. Since we use a symmetrical Gaussian function to model the PSF, the mean-squared positional error is given by:

$$\sigma_{x,y}^2 \approx \frac{s^2 + \frac{a^2}{12}}{N_m} + \frac{4\sqrt{\pi}s^3 b_m^2}{a N_m^2},$$

where $s$ is the standard deviation of the PSF, $a$ is the pixel size in the image, $N_m$ is the total number of photons measured from the molecule $m$, and $b_m$ is the number of background photons measured in the localization window (*Thompson et al., 2002*). We calculate the photon conversion factor for our camera by measuring the mean and variance of the camera response counts as a function of illumination intensity (*Newberry, 1998*). The mean localization precision was 12 nm (*Figure 2—figure supplement 1*).

## Two-color experiments

Two-color super-resolution imaging was performed with dSTORM and STORM, each of which has advantages and limitations for this particular application. Briefly, for two-color dSTORM imaging of Nup358 and importin-β, an antibody against Nup358 (abNup358) was labeled with either ATTO520 or Alexa405, and importin-β was labeled as in single-color dSTORM imaging. For two-color STORM imaging, abNup358 was labeled with both Cy3 and Cy5 (in an approx. 4:1 ratio), and importin-β was labeled with Alexa405 and Cy5. Specific activation of these dyes was performed by providing a short pulse (~1 frame) of the appropriate 405 nm or 532 nm laser at low power.

Correct labeling of the two proteins was confirmed using widefield imaging. Activation with 532 nm and imaging with a 640 nm laser showed bright nuclear envelope staining, suggesting that Nup358 was indeed labeled correctly. Activation using the 405 nm laser also yielded a bright nuclear envelope, with significant protein localized within the nucleus; this staining pattern is typical of that observed in our confocal imaging of importin-β.

We found that it is difficult to perform two-color localization imaging while trying to localize two pools of protein (such as importin-β and Ab-Nup358) that are in close proximity (<50 nm). Standard two-color dSTORM suffers from chromatic errors, as the two fluorophores must have well-separated excitation profiles, which necessarily results in spectrally well-separated emission maxima and therefore rendering the experiment very sensitive towards chromatic aberrations. The chromatic aberrations manifest as uncertainty in alignment and co-localization of molecules. STORM does not suffer from chromatic aberrations since both probe pairs use Cy5/Alexa647 as their reporter. However, in STORM, activation cross-talk must be considered—since both probe pairs emit the same color, it is no longer possible to assign the molecular identity of a signal with absolute certainty.

Given the various advantages and limitations of dSTORM and STORM, we judged it best to apply both methods to equivalent samples and then compare the results as a consistency check. As shown in *Figure 3—figure supplement 1*, imaging the equatorial plane (i.e., viewing the NPC from the side) the importin-β signal does not coincide with the Nup358 signal demarcating the cytoplasmic face of the NPC; rather the bulk of the imp-β signal was about 25 nm closer to the center of the nucleus, consistent with the imp-β being primarily located within the NPC. As a further crosscheck, we applied both methods also to the basal surface of the nucleus. As expected when imaging the 'front' of the

NPC (i.e., looking at it from the cytoplasm), the signals from the Nup358 and imp-β spatially merged when imaging the basal surface of the nucleus.

## STORM images

Display images were created using the STORM localization data by bin-sorting the data with an appropriate bin size. Let $K$ be the set of $n$ STORM localizations $\{x_1,...,x_n\}$. The data can be sorted into bins with size $h$ (typically the localization precision of our instrument) according to the following equation:

$$\mathbf{I}_k = \left\lfloor \frac{1}{h}\, \mathbf{x}_k \right\rfloor.$$

## NPC identification, extraction, and alignment

Individual NPCs were identified and extracted automatically, using a similar method to that used previously (*Lowe et al., 2010*). After identifying a closed path describing the nuclear envelope from widefield fluorescence and down-sampled dSTORM/STORM images, we use the interpolated surface normal vectors of this path to position a sliding window normal to the envelope at positions along the envelope path. We count the number of localizations found within the window at each position along the envelope path. This 'envelope histogram' contains distinct regions containing high numbers of localizations, which correspond to the centroids of NPCs.

Having located the NPCs in the envelope, we use the interpolated surface normal to rotate the importin-β localizations corresponding to a single NPC into a common frame, whereby the cytoplasm–nucleus vector is oriented vertically down. Next, we use cross-correlation and reference-free alignment, to align each NPC image with sub-pixel resolution. Image clustering is used to identify groups of structures within the data set. Once all NPC structures are correctly aligned, we can calculate statistics including axial distributions, mean NPC images and positional variance maps.

## Single molecule counting

The stepwise-photobleaching method was adapted from the approach of *Leake et al. (2006)*. The stepwise-photobleaching method relies on the irreversible and stochastic bleaching of fluorescent proteins upon repeated exposure. The sample is illuminated with an excitation light intensity low enough to slowly bleach it until background emission is reached. Plotting the intensity of a spot of interest over time results in an exponential decay function. Ideally, this function contains discernible steps. Each step corresponds to a bleaching event of a single molecule. When the number of molecules to count increases, the chance of having several bleaching steps at the same time also increases, resulting in steps having sizes that are multiples of a single bleach step. We added mCherry tagged impβ to permeabilized cells, fixed them, and imaged the basal envelope of the nuclei. The angle of the beam was chosen in such a way to minimize the background (normally just slightly higher than the optimal TIRF angle). The desired focal plane was found using the lowest laser power possible (~150 μW) to avoid bleaching during focusing. As soon as the correct focal plane was found the laser power was increased to about 3 mW and movie recording was started. Movies were acquired at 50 Hz and normally have a length of about 180 s (or until background emission was reached). Four different conditions were tested: WT +Ran+GTP, WT −Ran−GTP, $R_{70\%}$Nup153 +Ran+GTP, $R_{70\%}$Nup153 −Ran−GTP.

## Cell preparation

Digitonin-permeabilized HeLa cells were treated with mCherry-impβ (0.5 μM), RanGDP (5 μM), NTF2 (4 μM), and an energy regenerating system (2 mM GTP, 0.1 mM ATP, 4 mM creatine phosphate, and 20 μ/ml creatine kinase) in transport buffer. The experimental mix was incubated at room temperature for 20 min before the cells were fixed with a 4% PFA solution for 15 min and washed 3 × 2 min with PBS. Energy deficient cells were prepared as above, but lacked RanGDP, NTF2, and the energy regenerating system. Nup153-knockdown cells were transfected ~48 hr before applying the protocol above.

## Microscope set-up

We used HILO illumination (*Tokunaga et al., 2008*). Other microscope details:

Microscope: Olympus IX81 stage, TIRF module.
Objective: Olympus UApoN 100XOTIRF, NA: 1.49.
Laser: Coherent Compass 561, 50 mW, fiber coupled.
Filter Cube: Semrock BrightLine mCherry-40LP-A.

Excitation Filter: 560/55, Dichromatic Mirror: 590LP, Emission Filter: 600LP.

Camera: Andor iXon 897.

A circular, step-variable neutral density filter was used to adjust the laser power between focusing and bleaching.

## Analysis code platform

The main analysis was written in *Matlab* R2012b (Mathworks). Spot detection was done using ImageJ.

## Detecting the pore

Initial spot detection was done on an image averaged over 200 frames. The average image was treated with simple convolutions to achieve spatial bandpass filtering. We convolved the original image with a Gaussian, creating a lowpassed version. Convolving the original image with a boxcar function created a second lowpass image. Subtracting the boxcar filtered from the Gaussian filtered image resulted in a bandpass filtered image. This method not only suppresses noise (via an appropriate Gaussian kernel) but also sharpens the object of interest (by adjusting the size of the boxcar kernel) (*Figure 4—figure supplement 1B*). Peak detection was done on the bandpassed image using the *ImageJ* plug-in *FindMaxima*. All local maxima were identified. For each maximum a flood-filling algorithm with a threshold gray level was performed. Maxima that had a previously filled area were discarded. For cases where several points had the same highest value inside the flood-filled area, the pixel closest to its geometric center was used. The output of the peak detection step was a binary image containing the locations of the maxima (or the geometric center in the case of multiple neighboring pixels). All spots detected outside the nucleus were removed by manually constraining the analysis to the nucleus (*Figure 4—figure supplement 1C*).

## Creating the time traces

The coordinates of the points detected in the previous step were extracted and saved to a structure array. For each region of interest a mask containing only its center point was created. These masks were then dilated with a three-pixel radius disk to create a region covering a typical nuclear pore. The time trace was then created for each pore by summing the pixel values inside the ROI for each frame of the raw movie. The raw signal was filtered with the edge-preserving Chung-Kennedy filter (*Chung and Kennedy, 1991*) (*Figure 4—figure supplement 1D*). All pairwise differences in this filtered trace $I(t)$ were calculated, that is, each $\Delta I_{xy} = I(t_x) - I(t_y)$, where $t_x > t_y$. For performance purposes, this part of the analysis was implemented in *C*. The distribution of the pairwise differences was calculated through a histogram with 500 bins. This pairwise differences distribution (PDD) ideally has periodical peaks separated by the most probable step size of the fluorophore (*Figure 4—figure supplement 1E*). This distribution was then smoothed and the individual peaks were extracted using a custom Matlab function. The mean differences between the peaks give a first approximation on the single-step size. To find the correct step size more accurately a power spectrum of the smoothed PDD was calculated using Matlab's *pwelch* function.

Peaks in the power spectrum were extracted and compared to the previously obtained rough estimate. The peak having the smallest absolute distance to the estimate was chosen. We now had the most probable single step $x$ for each of the bleaching traces. The calculation of the number of molecules bleached in this particular spot was then straightforward: $N = \frac{I_{initial} - I_{final}}{x}$, where $I_{initial}$ is the initial intensity defined as the mean value of the first 10 samples. $I_{final}$ is the final intensity, given by the mean of the last 1000 points of the trace. Each nucleus gives usable information of about 80–200 pores. For each condition, 10–15 movies were recorded, yielding several thousand pores per condition. All counts for a given condition were combined to generate statistics and histograms (*Figure 4—figure supplement 1*).

## Mann–Whitney confidence intervals of the differences

We used a confidence interval based on the Mann–Whitney distribution (*Conover, 1999*) to generate *Table 1*. The Mann–Whitney test does not make any assumptions on the shape of the distributions. It only requires both samples to be random samples from their respective populations. Consider two populations having identical but unknown distribution functions: $X_1,...,X_n$ and $X_1,...,X_n$. A parameter $k$ can be calculated as follows: $w_{\alpha/2} - n\frac{n+1}{2}$ where $w_{\alpha/2}$ is the $\frac{\alpha}{2}$ quantile for $n$ and $m$. The quantile of the Mann–Whitney distribution can be approximated using the quantile $z_p$ of the normal distribution: $w_p = \frac{n(N+1)}{2} + z_p \sqrt{\frac{nm(N+1)}{12}}$, where $N = n + m$. Next, all possible pairs $(X_i, Y_j)$ are created. The $k$ largest

and $k$ smallest differences are extracted. The upper limit U is defined as the $k$th largest difference. The lower limit L is defined as the $k$th smallest difference. The final confidence interval is then given by $P[L \leq E(X) - E(Y) \leq U] \leq (1 - \alpha)$. The calculation of the confidence interval was implemented in Matlab.

## In vitro gelation; fluorescence fluctuation spectroscopy

### Buffer conditions of the in vitro gelation study

Recombinantly-expressed impβ-YFP and the FG domain of Nup153 (amino acids 874–1475) were mixed together in 500 µl of PBS with one or more of the following factors at final concentrations of 50 nM impβ-YFP, 0.5 µM Nup153FG, 2 µM Ran•GDP, 2 µM RanQ69L•GTP, 100 nM TRN1-GFP, and 2 mM DTT in a glass-bottomed Lab-Tek chamber (Nunc). Samples were prepared at room temperature and analyzed either immediately after preparation (for analyzing kinetics of complex formation) or after 30 min.

### Microscope setup and analysis of the fluctuation traces

Fluorescence fluctuation spectroscopy measurements were performed on a home-built apparatus based on a Nikon TE2000-E inverted fluorescent microscope as described previously (*Forstner et al., 2006*). A 485 nm pulsed diode laser (PicoQuant, Berlin, Germany) were used to excite the sample. The laser beam was coupled into an optical fiber and focused by a 100× TIRF objective (Nikon) into the sample to excite the fluorescent probes. The emission light was filtered by a notch filter and a 50-µm confocal pinhole, followed by a short-pass (550 nm) color filter before directing toward the avalanche photodiodes (APDs) (Perkin Elmer, Canada). The photon arrival time was recorded and processed with a hardware correlator (Correlator.com). Igor (WaveMetrics Inc., Portland, OR) was used to analyze the photon counting histograms described previously (*Muller et al., 2000*; *Xu et al., 2011*) with a bin time of 100 µs. Confocal volume was calibrated by 100 nM of Alexa-488 in room temperature with known diffusion coefficient (D = 430 µm$^2$/s) (*Nitsche et al., 2004*). Samples that were directly compared were done on the same day to minimize differences in the instrument settings.

### Confocal imaging of large impβ•Nup153FG complexes

Impβ-YFP (4 µM) and Nup153FG (5 µM) were incubated together at room temperature and formed large, micron-sized structures on a timescale of minutes (*Figure 6A*). These structures resembled aggregated protein conglomerates that were massive enough to settle to the surface due to gravity.

## Acknowledgements

This work was partially supported by the NIH/NIGMS (R01GM077856 to JTL and R01GM058065 to KW), the NIH/NCI (NCI U54CA143836 to JTL and JTG), and NIH/NIAID PO1AI091580 to JTG. We thank Phillip Jess for construction of the real-time FRAP microscope, Will Draper for additional Python microscope control software, Ann Fischer and the UC Berkeley tissue culture facility for cell culture, and the UC Berkeley MacroLab for protein purification. We thank Ulrike Kutay for discussions and for critically reading the manuscript and Roderick Lim and Naoko Imamoto for plasmids.

## Additional information

### Competing interests

KW: Reviewing editor, *eLife*. The other authors declare that no competing interests exist.

### Funding

| Funder | Grant reference | Author |
| --- | --- | --- |
| National Institute of General Medical Sciences (NIGMS) | R01GM077856 | Jan T Liphardt |
| National Institute of General Medical Sciences (NIGMS) | R01GM058065 | Karsten Weis |
| National Cancer Institute (NCI) | U54CA143836 | Jay T Groves, Jan T Liphardt |

| Funder | Grant reference | Author |
|---|---|---|
| National Institutes of Health (NIH) | U54CA143836 | Jay T Groves, Jan T Liphardt |
| National Institute of Allergy and Infectious Diseases (NIAID) | PO1AI091580 | Jay T Groves |
| National Institutes of Health (NIH) | R01GM077856 | Jan T Liphardt |
| National Institutes of Health (NIH) | R01GM058065 | Karsten Weis |
| National Institutes of Health (NIH) | PO1AI091580 | Jay T Groves |

The funders had no role in study design, data collection and interpretation, or the decision to submit the work for publication.

### Author contributions
ARL, JHT, Conception and design, Acquisition of data, Analysis and interpretation of data, Drafting or revising the article; JY, MG, WYCH, Acquisition of data, Analysis and interpretation of data; JTG, Analyzed the fluorescence fluctuation data; KW, JTL, Conception and design, Analysis and interpretation of data, Drafting or revising the article

### Author ORCIDs
Karsten Weis, http://orcid.org/0000-0001-7224-925X

## Additional files

### Supplementary file
• Supplementary file 1. Plasmid constructs.

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
