## [Decision Letter]

Thank you for sending your work entitled “Importin-β Modulates the Permeability of the Nuclear Pore Complex in a Ran-dependent Manner” for consideration at *eLife*. Your article has been favorably evaluated by Vivek Malhotra (Senior editor and Reviewing editor) and 2 reviewers.

The Reviewing editor and the other reviewers discussed their comments before we reached this decision, and the Reviewing editor has assembled the following comments to help you prepare a revised submission.

1) Show raw data for the Imp-YFP experiment that is a part of Figure 1.

2) Figure 3. Display the raw data. It is very crucial that you explain how the data was generated and normalized.

3) Figure 4. This figure also suffers from suboptimal presentation. It should be revised and clarified for the readers.

4) Figure 6. Show the exact concentration of the reagents for the procedures used and include an experiment to test the effect of Ran on Nup153G-tranportin complex. This is an important control and will help rationalize the data on Nup153-Impbeta.

The detailed comments of the reviewers follow.

*Reviewer #1*:

The work by Lowe et al. uses a set of fluorescence assays on permeabilized cells to study the permeability barrier of the NPC. A major conclusion from this work is that Ran itself can modulate directly the permeability barrier, by modulating two (novel) Imp-beta populations that are located at the NPC. They use siRNA knockdowns and in vitro binding studies, to argue that this modulation can occur via Nup153. The manuscript is rich in experiments, and covers many interesting aspects that will appeal to a broad readership. There are several places though, that I find confusing, or where simple experiments to rule out alternative conclusions have not been made. I will go through these partially minor, partially severe concerns in the order the Figures are presented in the paper.

Figure 1: Panel A-C) visualize a trivial concept, that cargo distribution changes once transport can be completed. What I find is missing is the raw data for the Imp-YFP experiment, which should be added somewhere. The observations in panel D-E) form the basis for many of their conclusions, and it is good to see that two different methods (FRAP, and photoconversion based FLAP) support their conclusion. With Ran, there is an enriched fast equilibrating pool of Imp-beta. This effect is then analyzed in Figure 2 with STORM based superresolution.

Figure 3: The claimed resolution of 12 nm is very high, and the analysis of positions provided makes this in principle credible (see below as well). What is clearly visible is that also two spatially separated populations of labelled Imp-beta along the transport axis can be found. This is certainly a very fascinating result, which also forms the basis for their conclusions. Convincingly (based also on the Nup358 analysis), one populations seems to be rather cytoplasmic, while the other is more nucleoplasmic. However, I have also some trouble following all conclusion and interpretations on this figure, as well as with some of the raw data and how it is displayed. In A) left panel, the Imp-beta is shown saturated (white color in the glow over low under representation), and the difference between the plus/minus Ran case is not that easily visible from the images (brightness is very different, and this is confusing, see below). The line profiles or projections (whatever it is) show a bigger difference, but conceptually I do not think this effect is that strong, that all types of definitive conclusions can be drawn from this. The apparent number of localizations went down, in line with Figure 1, but distribution change is less obvious in the raw data. This is also particular notable with respect to the data displayed in panel E and G. Here, the color coding show abundance levels, but in the white drawn box in (F), there are more abundant species (more yellow and red) than in the same box in E. This seems quite in contrast to the image panels shown in A. How was the data normalized and displayed? The difference between panel plus/minus Ran could also arise from different localization precision in the tow measurements. dSTORM is sensitive to high concentrations of dye, as multiple emitters could get activated, and then in the higher concentrated Imp-beta case, the clear bimodal distribution could get blurred. This technical issue could also explain the minor difference between the plus/minus Ran case. I do agree however, that in the Nup153 knockdown the distribution is clearly different. What does surprise me though is that I would have expected to see much less Imp-beta in the delta-Nup153 case, in line with the counting results from Figure 3. However, the image here looks at least as bright as the Imp-beta minus Ran case. Either the authors were not careful with normalizations/averaging or contrast/brightness adjustments, or something odd is going on here in the entire Figure 3. In the latter case, this would be a concern for a larger set of conclusions drawn in this paper.

Also, only in panel A) the localization data for Imp-beta no Ran and delta Nup153 seems to smear substantially into the nucleoplasm, while the nucleocytoplasmic fraction of Imp-beta should be enriched in particular in the Imp-beta plus Ran case, why?

Figure 4: While I do believe that overall reproducibility may be sufficient to make comparative analysis, I discourage of giving specific numbers in the text with only using the “approximate” sign, as true numbers might be off several fold. Stepwise bleaching analysis suffers from many problems that make determination of absolute abundance levels very, very difficult.

A) Optical sectioning in HILO is not sufficient enough, to have no contribution from out of focus NPCs in the field of view.

B) YFP, can go into dark states and return spontaneously, as every FP (leading to overcounting when just counting bleaching steps). Even conversion of YFP into CFP has been debated in a set of Nature Methods papers between 2005-2006.

C) Maturation time of FP is also an issue, that can further yield wrong numbers. Corrections for all those parameters were not done, and are also difficult to do. A few procedures on how to do this have been established by Annibale et al. and Lee et al. e.g. for photoswitchable proteins, because traditional FPs have all these issues, of which the contribution is much harder to estimate then for well characterized real “switchable” proteins.

D) Due to crowding, Homo FRET might affect overall brightness, when ∼ 70 YFP get squeezed onto a small volume of an NPC transport channel. The effect cannot be easily estimated, and can contribute to substantial imprecision. An “approximate sign” does not seem to sufficiently account for this in my eyes.

Again, the overall conclusion might not be affected, but the true number of Imp-beta at the pore might easily be several folds off. This should be clarified for the reader, as otherwise they simplify a big problem in quantitative proteomics: Getting abundance levels from fluorescence images is not as easy as just bleaching the sample, especially not for high abundant proteins, where the initial brightness is very high.

Figure 6: I have trouble following the logic of the motivation and description of these experiments in relation to the in vivo experiments, while at the same time I find the result intriguing, which is, that the Importin beta cross links Nup. Overall, this does not come very much as a surprise, since Imp-beta has several binding sites for FG repeats and this will lead to aggregation (the authors need to report the exact concentration in the methods part, and not just “micromolar for both”). I assume, the same experiments can be shown to work with most transport receptors of this type and larger FG Nups. I would speculate, transportin will give the same effect, and they could easily do such an experiment. If also Ran can solubilize Nup153FG-transportin complexes, I wonder how such a generic mechanisms can be rationalized in line with i) their finding that in vivo the Imp-beta and transportin are doing different things (and how this in vitro experiment is then supporting their conclusions) ii) with their model of specific relevance's of certain proteins (Nup153 and Imp-beta).

Much of my criticism can be addressed with simple experiments that could be performed in a timely manner, and should thus be possible within a major revision.

Minor comments:

As a minor comment, in the Discussion, they speculate also about the potential role of Imp-beta structure, in light of a similar architecture of some scaffold Nups. How do they rationalize this, since transportin is also similar in structure, but is apparently not doing the same as Imp-beta.

Also, with Figure 6 and Figure 7, the authors use the words “reminiscent” in terms of “gels”, a topic of particular relevance in the transport field. Besides not every aggregate necessarily also being a gel (their structures could be, but this would need different assay to confirm) their potential gels are conceptually very different than the ones originally introduced by the Gorlich lab. In their gels, the Nup is crosslinked by Imp-beta, while the Gorlich gels is based on homotypic interactions between Nups only. I think using the term “reminiscent” as used by the authors is not pointing enough to the fundamental difference of their model. It also remains open, to what extend Nup153 is the only Nup that would cause such an effect. Other factors might certainly be important as well, but naturally, not everything can be addressed in a single study. E.g., a previous single molecule study that has directly shown that Nup50 is important for cargo dissociation of Importin complexes is neither cited nor discussed.

*Reviewer #2:* Nuclear pore complexes mediate traffic between the nucleus and the cytoplasm. The selectivity and efficiency of transport through the nuclear pore complex is not fully understood. In current models, Ran in its GTP or GDP bound form gives direction to transport by modulating the interaction of Impβ with its cargo's on either side of the NPC. The current manuscript provides proof that RanGTP influences the permeability of the NPC itself. Based on modern microscopy techniques that reveal the number, localization and mobility of Impβ molecules in the NPC the authors propose that impβ and Nup153 interact at the nuclear side of the NPC where they form a Ran-regulated meshwork. Transport experiments show that modulation by Ran alters both passive and active transport.

The manuscript thus presents new interesting insights into the transport mechanism of the NPC. Future studies will have to resolve if in vivo modulation of the Ran gradient is an important mechanism to fine-tune NPC permeability; it is an exciting possibility.

I find this an exciting an impressive piece of work on an important topic that should be accepted for publication in *eLife*.

---

## [Author Response]

We thank both reviewers for their excellent points. The central new experiment in this manuscript is the requested exploration of the effect of Ran on a hypothesized Nup153FG-tranportin complex. Despite the apparent simplicity of the experiment – consisting of replacing impβ with transportin in the fluctuation correlation spectroscopy measurements – it ultimately took 5 months to learn how to purify sufficiently large amounts of transportin in a pure enough form to actually perform those measurements. We apologize for this delay. We now report the outcome of those measurements in Figure 6.

Briefly, when mixed, Nup153FG and transportin form a material, just like when Nup153FG is mixed with imp-β. Critically, however, the Nup153FG•transportin material is not dissolved by Ran, pointing to a key functional difference between the Nup153FG•imp-β and Nup153FG•transportin materials. Further, the passive permeability experiments show that only the imp-β mediated barrier is able to restrict the movment of 2xGFP. These in vitro and in nuclei biophysical results therefore support and round-out the previous literature suggesting that:

1) transportin-mediated import has different Ran requirements compared to imp-β-mediated transport (Ribbeck, K., et al., The translocation of transportin-cargo complexes through nuclear pores is independent of both Ran and energy*.* Current Biology, 1999. 9(1): p. 47-50),

2) transportin interacts with Nup153 at different sites than does imp-β (Shah, S. and D.J. Forbes, Separate nuclear import pathways converge on the nucleoporin Nup153 and can be dissected with dominant-negative inhibitors*.* Current Biology, 1998. 8(25): p. 1376-1386)), and,

3) imp-β may have a special role within the NPC (e.g. Kapinos, L. E., Schoch, R. L., Wagner, R. S., Schleicher, K. D., & Lim, R. Y. H. Karyopherin-centric control of nuclear pores based on molecular occupancy and kinetic analysis of multivalent binding with FG nucleoporins. Biophys J 106, 1751-1762 (2014).

We emphasize that materials (such as meshes and gels) can have an extremely broad range of biophysical properties, such as effective pore size, gelation temperature, gelation kinetics, hetero- vs. homotypic interactions, effect of ionic strength, critical crosslinker concentrations etc. In our case, the experiments raise the possibility that the Nup153FG•transportin material has a larger effective pore size than the Nup153FG•imp-β material, and the experiments show that imp-β and transportin differ significantly in terms of their interactions with Ran.

We hope you like the revised manuscript and we again apologize for the time it took to complete the transportin fluctuation correlation spectroscopy study (new Figure 6).

Reviewer 1:

*The work by Lowe et al. uses a set of fluorescence assays on permeabilized cells to study the permeability barrier of the NPC. A major conclusion from this work is that Ran itself can modulate directly the permeability barrier, by modulating two (novel) Imp-beta populations that are located at the NPC. They use siRNA knockdowns and in vitro binding studies, to argue that this modulation can occur via Nup153. The manuscript is rich in experiments, and covers many interesting aspects that will appeal to a broad readership*.

Thank you for your endorsement and the many excellent points you raise.

*There are several places though, that I find confusing, or where simple experiments to rule out alternative conclusions have not been made. I will go through these partially minor, partially severe concerns in the order the Figures are presented in the paper*.

Figure 1*: Panel A-C) visualize a trivial concept, that cargo distribution changes once transport can be completed. What I find is missing is the raw data for the Imp-YFP experiment, which should be added somewhere. The observations in panel D-E) form the basis for many of their conclusions, and it is good to see that two different methods (FRAP, and photoconversion based FLAP) support their conclusion. With Ran, there is an enriched fast equilibrating pool of Imp-beta. This effect is then analyzed in*
Figure 2
*with STORM based superresolution*.

The panels may be trivial (and completely consistent with previous literature) but nonetheless we think it is very important to show them, if only for the more general reader who is not completely familiar with the NPC field. We have added the raw data for the Imp-YFP experiment to panel B of the figure, as well as intensity line profiles to show the absolute levels of Imp-YFP and cargo in the respective compartments. The figure legend has also been updated accordingly.

Figure 3*: The claimed resolution of 12 nm is very high, and the analysis of positions provided makes this in principle credible (see below as well). What is clearly visible is that also two spatially separated populations of labelled Imp-beta along the transport axis can be found. This is certainly a very fascinating result, which also forms the basis for their conclusions. Convincingly (based also on the Nup358 analysis), one populations seems to be rather cytoplasmic, while the other is more nucleoplasmic. However, I have also some trouble following all conclusion and interpretations on this figure, as well as with some of the raw data and how it is displayed. In A) left panel, the Imp-beta is shown saturated (white color in the glow over low under representation), and the difference between the plus/minus Ran case is not that easily visible from the images (brightness is very different, and this is confusing, see below). The line profiles or projections (whatever it is) show a bigger difference, but conceptually I do not think this effect is that strong, that all types of definitive conclusions can be drawn from this. The apparent number of localizations went down, in line with*
Figure 1*, but distribution change is less obvious in the raw data. This is also particular notable with respect to the data displayed in panel E and G. Here, the color coding show abundance levels, but in the white drawn box in (F), there are more abundant species (more yellow and red) than in the same box in E. This seems quite in contrast to the image panels shown in A. How was the data normalized and displayed? The difference between panel plus/minus Ran could also arise from different localization precision in the tow measurements. dSTORM is sensitive to high concentrations of dye, as multiple emitters could get activated, and then in the higher concentrated Imp-beta case, the clear bimodal distribution could get blurred. This technical issue could also explain the minor difference between the plus/minus Ran case. I do agree however, that in the Nup153 knockdown the distribution is clearly different. What does surprise me though is that I would have expected to see much less Imp-beta in the delta-Nup153 case, in line with the counting results from*
Figure 3*. However, the image here looks at least as bright as the Imp-beta minus Ran case. Either the authors were not careful with normalizations/averaging or contrast/brightness adjustments, or something odd is going on here in the entire*
Figure 3*. In the latter case, this would be a concern for a larger set of conclusions drawn in this paper*.

*Also, only in panel A) the localization data for Imp-beta no Ran and delta Nup153 seems to smear substantially into the nucleoplasm, while the nucleocytoplasmic fraction of Imp-beta should be enriched in particular in the Imp-beta plus Ran case, why*?

Your excellent points have helped us to greatly clarify and simplify the figure and language. Localisation microscopy data are notoriously hard to convey in a concise, systematic and unbiased way, whilst simultaneously acknowledging the caveats to the method as referenced in the reviewer’s comments. We went through every line of the processing code and we found an arithmetic error that affected the normalization of the image displayed in the Nup153 subpanel, which resulted in an incorrect scaling of the brightness compared to the other conditions. This error did not affect panel C, since all of those curves are true probability distribution functions. We understand how confusing this was – we apologize! We have clarified the figure in the following ways.

1) We fixed the arithmetic error.

2) We have rewritten the legend.

3) A new figure supplement showing the ‘raw’ localization data has been added. In panel B, this figure shows thirty localization datasets for single NPCs for each of the three conditions (-RanGTP, +RanGTP, Δ153_70%_). These are not images in the traditional sense, but come close to something that one would want to compare to a confocal image. Indeed, they recapitulate what is seen in the confocal images (Figure 1).

4) A new figure supplement showing the number of counts for each condition has been added. These are just raw numbers. It shows the differences in abundance for each of the three conditions, acknowledging the caveats to calculating absolute numbers by localization microscopy, as referenced up by the reviewer.

5) We have deleted the original subpanel D and simplified the figure. Briefly, Figures 3C and D (formerly E-F) show probability distribution functions, i.e. their integrals all been set to have area = 1. Therefore, any overall changes of counts among the various conditions are removed, by design, giving true 1D probability distribution functions. The goal of Figure 3 is to present spatial changes of imp-β arrangement. As can be seen, the imp-β only condition has a relatively flat peak and, relative to that peak, has significant imp-β signal at the nucleoplasmic face of the pore. When Ran is added, we are dealing with an ‘active’ transport competent pore. The signal in the middle of the pore is now clearly bimodal and there is depletion of imp-β signal at the nucleoplasmic face of the pore. Since these are all true probability distribution functions (whose area is always = 1), the PDF in the middle rises to compensate for the loss of signal at the nucleoplasmic side of the pore. When Nup153 is reduced, the signal from the entire nucleoplasmic side of the pore drops, leaving only one shouldered peak, which becomes the predominant feature of the PDF.

Figure 4*: While I do believe that overall reproducibility may be sufficient to make comparative analysis, I discourage of giving specific numbers in the text with only using the* “*approximate*” *sign, as true numbers might be off several fold. Stepwise bleaching analysis suffers from many problems that make determination of absolute abundance levels very, very difficult*.

*A) Optical sectioning in HILO is not sufficient enough, to have no contribution from out of focus NPCs in the field of view*.

*B) YFP, can go into dark states and return spontaneously, as every FP (leading to overcounting when just counting bleaching steps). Even conversion of YFP into CFP has been debated in a set of Nature Methods papers between 2005-2006*.

*C) Maturation time of FP is also an issue, that can further yield wrong numbers. Corrections for all those parameters were not done, and are also difficult to do. A few procedures on how to do this have been established by Annibale et al. and Lee et al. e.g. for photoswitchable proteins, because traditional FPs have all these issues, of which the contribution is much harder to estimate then for well characterized real* “*switchable*” *proteins*.

*D) Due to crowding, Homo FRET might affect overall brightness, when ∼ 70 YFP get squeezed onto a small volume of an NPC transport channel. The effect cannot be easily estimated, and can contribute to substantial imprecision. An* “*approximate sign*” *does not seem to sufficiently account for this in my eyes*.

*Again, the overall conclusion might not be affected, but the true number of Imp-beta at the pore might easily be several folds off. This should be clarified for the reader, as otherwise they simplify a big problem in quantitative proteomics: Getting abundance levels from fluorescence images is not as easy as just bleaching the sample, especially not for high abundant proteins, where the initial brightness is very high*.

We have re-plotted Figure 4 to simplify presentation of the experimental procedure and the results of the protein counting by stepwise photo-bleaching experiments. Whilst we agree with the reviewer that there are difficulties in reporting absolute numbers of proteins (due to some of the reasons mentioned), we note that our numbers are similar to those reported with an independent (albeit also fluorescence based) imaging method (59). This group found 110±50 impβ-GFP per pore in the absence of RanGTP. We have clarified the text to make reference to the difficulties and potential sources of error in stating absolute numbers, and emphasized that it is *the change in abundance* under different conditions that is important to the conclusions we draw. We have also removed “absolute” numbers from the abstract.

Figure 6*: I have trouble following the logic of the motivation and description of these experiments in relation to the in vivo experiments, while at the same time I find the result intriguing, which is, that the Importin beta cross links Nup. Overall, this does not come very much as a surprise, since Imp-beta has several binding sites for FG repeats and this will lead to aggregation (the authors need to report the exact concentration in the methods part, and not just* “*micromolar for both*”*). I assume, the same experiments can be shown to work with most transport receptors of this type and larger FG Nups. I would speculate, transportin will give the same effect, and they could easily do such an experiment. If also Ran can solubilize Nup153FG-transportin complexes, I wonder how such a generic mechanisms can be rationalized in line with i) their finding that in vivo the Imp-beta and transportin are doing different things (and how this in vitro experiment is then supporting their conclusions) ii) with their model of specific relevance's of certain proteins (Nup153 and Imp-beta)*.

We have now carried out the suggested fluorescence fluctuation spectroscopy experiments with transportin (TRN1). We again detected spikes in intensity upon addition of Nup153FG indicating the formation of large complexes (new Figure 6). As noted by the reviewer, this is not surprising given that TRN1 is structurally similar to impβ and likely also contains multiple FG-binding sites. Remarkably, and in stark contrast to impβ, the RanQ69L•GTP had no observable effect on the complexes, even though TRN1 and RanGTP are known binding partners. We thank the reviewer for suggesting this important experiment as it strengthens our central argument that there is a particular Ran-modulated interaction between impβ and Nup153. Also, the exact experimental conditions and protein concentrations used for these experiments are now detailed in the main text and Methods section.

*Much of my criticism can be addressed with simple experiments that could be performed in a timely manner, and should thus be possible within a major revision*.

*Minor comments*:

*As a minor comment, in the Discussion, they speculate also about the potential role of Imp-beta structure, in light of a similar architecture of some scaffold Nups. How do they rationalize this, since transportin is also similar in structure, but is apparently not doing the same as Imp-beta*.

Yes, impβ and TRN1 do not appear to be functionally equivalent in the NPC despite both being structurally similar to each other and to certain scaffold nucleoporins. This may be partly explained by their differential behaviors in response to RanGTP-binding as now shown in Figure 6. We also now highlight a previous study showing that impβ and TRN1 bind distinct regions on Nup153 (52) that may explain their different behaviors in the NPC. Furthermore, the dissociation constants of transport receptors to FG nucleoporins have been observed to vary significantly, from an overall K_D_ ∼ 4 µM for TRN1 binding to the NPC (Ribbeck *et al.*, 2001) to K_D_’s ranging from 1.7 µM to 0.4 nM for impβ binding to various FG nucleoporins (Pyhtila *et al.*, 2003). These differences in binding affinities likely affect the degree to which these karyopherins can interact with FG nucleoporins within the pore. Summarizing, the new experiments reinforce the notion that structural similarity between two proteins does not guarantee their functional equivalence.

*Also, with*
Figure 6
*and*
Figure 7*, the authors use the words* “*reminiscent*” *in terms of* “*gels*”*, a topic of particular relevance in the transport field. Besides not every aggregate necessarily also being a gel (their structures could be, but this would need different assay to confirm) their potential gels are conceptually very different than the ones originally introduced by the Gorlich lab. In their gels, the Nup is crosslinked by Imp-beta, while the Gorlich gels is based on homotypic interactions between Nups only. I think using the term* “*reminiscent*” *as used by the authors is not pointing enough to the fundamental difference of their model*.

This is great point that cannot be emphasized enough. We agree that the nature of the cross-linking (homotypic FG interactions versus impβ mediated) is one critical difference between our material and that of the Gorlich gels. Another critical difference is that our materials form (1) in physiological pH and ionic strength, (2) they form rapidly, (3) and the Nup153FG•imp-β material is sensitive to Ran, which can dissolve the material. We have tried to sharpen the language. In general, we also tried to stay away from using the word ‘gel’, since from a formal rheological perspective, the only way to actually confirm that we are dealing with a classical gel would be though measurement of the material’s dynamic loss modulus, which would require the material to be prepared in bulk quantities and subject to dynamic mechanical testing.

*It also remains open, to what extend Nup153 is the only Nup that would cause such an effect. Other factors might certainly be important as well, but naturally, not everything can be addressed in a single study. E.g., a previous single molecule study that has directly shown that Nup50 is important for cargo dissociation of Importin complexes is neither cited nor discussed*.

We’ve added a paragraph to the manuscript that references the Musser paper although this is an open question and we are uncomfortable discussing guesses about other Nups. We do note, however, that not all Nups are functionally identical and that some Nups are clearly more important to cell viability than others. In the case of Nup153, complete deletion results in loss of cell viability, which is consistent with Nup153 having a special, fundamental role in cell physiology, e.g. by interacting with other proteins and Ran in such a way as to allow cargos to be reliably and actively imported into the nucleus. That said, we would not be surprised if a small number of other Nups also turned out to exhibit particular biophysical capabilities that are modulated by Ran.